# Air-sea gas exchange in a seagrass ecosystem — results from a $^3$He/SF$_6$ tracer release experiment

Ryo Dobashi[1], David T. Ho[1]

[1]Department of Oceanography, University of Hawai'i at Mānoa, 1000 Pope Road, Honolulu, Hawaii 96822, USA

*Correspondence to*: Ryo Dobashi (rdobashi@hawaii.edu)

**Abstract.** Seagrass meadows are one of the most productive ecosystems in the world and could help mitigate the increase of atmospheric $CO_2$ from human activities. However, understanding the role of seagrasses in the global carbon cycle requires knowledge of air-sea $CO_2$ fluxes and hence knowledge of the gas transfer velocity. In this study, gas transfer velocities were determined using the $^3$He and SF$_6$ dual tracer technique in a seagrass ecosystem in south Florida, Florida Bay, near Bob Allen Keys (25.02663°N, 80.68137°W) between 1 and 8 April 2015. The observed gas transfer velocity normalized for $CO_2$ in freshwater at 20° C, $k(600)$, was 4.8 ± 1.8 cm h$^{-1}$, which was lower than that calculated from published wind speed/gas exchange parameterizations. The deviation in $k(600)$ from other coastal and offshore regions was only weakly correlated with tidal motion and air-sea temperature difference, implying that wind is the dominant factor driving gas exchange. The lower gas transfer velocity was most likely due to wave attenuation by seagrass and limited wind fetch in the study area. A new wind speed/gas exchange parameterization is proposed ($k(600) = 0.143u_{10}^2$), which might be applicable to other seagrass ecosystems and wind fetch limited environments.

## 1 Introduction

Seagrass meadows are one of the most productive ecosystems in the world and stock as much as 4.2–8.4 PgC in their soils (Fourqurean et al., 2012). Because some of the organic carbon produced by seagrasses are refractory and accumulate on the seafloor, seagrass meadows are expected to be blue carbon sinks that help mitigate the increase of anthropogenic $CO_2$. Seagrasses are estimated to bury 45−190 g C m$^{-2}$ yr$^{-1}$, a significantly higher rate compared to terrestrial forests (0.7−13.1 g C m$^{-2}$ yr$^{-1}$; Mcleod et al., 2011; Duarte et al., 2005). However, recently, the role of seagrasses in the global carbon cycle has been revisited, as carbon emissions from seagrasses were found to be large (Howard et al., 2017; Van dam et al., 2021; Schorn et al., 2021). Howard et al. (2017) examined the stock of organic and inorganic carbon in the soil of seagrass meadows in Florida Bay and southeastern Brazil and found that the soils in both regions have more inorganic than organic carbon. They suggested that both regions are sources of $CO_2$ to the atmosphere by assuming 0.6 mol of $CO_2$ is produced when 1 mol of CaCO$_3$ is

produced. Schorn et al. (2021) reported that the seagrasses in the Mediterranean Sea emit 106 µmol m$^{-2}$ d$^{-1}$ methane, mainly from their leaves.

Knowledge of the gas transfer velocity ($k$) is needed to understand the role of seagrass ecosystems in the global carbon cycle since air-sea $CO_2$ flux is a function of $k$ and the air-sea difference in the partial pressure of $CO_2$ ($pCO_2$). There are several methods to determine $k$ in the field. The $^3$He/SF$_6$ dual tracer technique, which we employed in this study, is a mass balance technique that involves injecting these tracers into the ocean and determining $k$ by measuring the change in the ratio of the two gases with time. The direct flux techniques, such as eddy covariance, measure the $CO_2$ flux in the air and $CO_2$ concentration both in the sea and air to derive $k$ (McGillis et al. 2001). The $k$ has also been estimated from the heat transfer velocity by assuming that the gas and heat transfer velocities are related by their diffusivities. However, the estimated gas transfer velocity from heat, $k_H$, has been found to overestimate the actual $k$ (e.g., Atmane et al., 2004).

Because $k$ is difficult to measure, it is often parameterized using easily and widely measured parameters such as wind speed. In deep offshore regions, wind is known to predict the gas transfer velocity well since wind creates waves and currents, which control turbulence and bubbles at the sea surface (Wanninkhof et al., 2009). Ho et al. (2018a) examined $k$ in Kaneohe Bay in Hawai'i and showed that $k$ can be estimated well by wind speed where the depth is deeper than 10 m. On the other hand, in shallow regions, other parameters become important as well (e.g., Ho et al., 2016; 2018b). Ho et al. (2016) showed that $k$ could be estimated well by wind speed and current speed in a shallow tidal estuary in south Florida because the current enhances bottom-generated turbulence. Ho et al. (2018b) examined $k$ in an emergent wetland where the depth < 1 m, and showed that $k$ can be parameterized by heat flux, rain rate, and current velocity. In the case of rain, rain rate is included in the parameterization because rainfall increases subsurface turbulence and $k$ (Ho et al., 1997a, 2000).

In Florida Bay, $k$ has been estimated from commonly used wind speed/gas exchange parameterizations. Zhang and Fischer (2014) determined the air-sea $CO_2$ flux to be 3.93 ± 0.91 mol m$^{-2}$ yr$^{-1}$ in Florida Bay; they used the wind speed/gas exchange parameterization determined from bomb-produced $^{14}$C inventory in the ocean by Wanninkhof (1992). Van Dam et al. (2020) estimated $k$ by using heat as a proxy ($k_H$) in Florida Bay and found that $k_H$ was lower compared with $k$ derived from published wind speed/gas exchange parameterizations when wind shear is relatively strong, even though $k_H$ is known to overpredict $k$. This finding suggests that previous wind speed/gas exchange parameterizations are not suitable for the seagrass-dominated area and a specific parameterization for these fetch-limited environments is needed. In the study presented here, we use a $^3$He/SF$_6$ tracer release experiment to determine $k$ in a shallow seagrass-dominated environment to understand processes that control $k$ and to derive a parameterization for this environment.

## 2 Methods

### 2.1 Study site

The $^3$He/SF$_6$ tracer release experiment was conducted between 1 and 8 April 2015 near Bob Allen Keys in Florida Bay (Fig. 1). Florida Bay is situated between the Everglades marsh and the Florida Keys in the southernmost part of Florida, USA,

and covers approximately 2,000 km$^2$. In this bay, the average depth is less than 3.5 m, and the vertical extent of seagrasses is between 0.08 and 0.2 m (Sogard et al., 1989). *Thalassia testudinum* and *Laurencia* are the dominant seagrass and macroalgae, respectively, in the benthic communities, with an average standing crop of 63.6 and 8.9 g dry weight m$^{-2}$, respectively (Zieman
et al., 1989). Seagrass density varies across the bay, and its standing crop is 0−20 g dry weight m$^{-2}$ in the summer around our study area (bottom figure in Fig. 1) (Zieman et al., 1989). The growth of seagrasses in Florida Bay is seasonal, with larger standing crops in spring and summer than in fall and winter (Zieman et al., 1999). The phytoplankton community is dominated by cyanobacteria, diatoms, and dinoflagellates (Philips and Badylak, 1996), with frequent cyanobacteria blooms in the central north region of the bay due to nutrient input from the land (Philips et al., 1999; Lavrentyev et al., 1998). Wind is persistent
from southeast to northwest during summer and from north to south during winter (Wang et al., 1994). The current speed is about 0.02−0.14 m s$^{-1}$ (Wang, 1998), and the tidal amplitude is 0.1−0.4 m (Wang et al., 1994).

## 2.2 Tracer injection and measurement

We injected $^3$He and $SF_6$ at a ratio of 1:340 into the water at the study location (25.0107°N, 80.692°W; black cross in Fig.
1) on 1 April 2015 for 1 minute via a length of diffuser tubing. After injection, we used an underway $SF_6$ analysis system (Ho et al., 2002) to measure surface water $SF_6$ concentrations every ~45 s. The system is composed of a gas extraction unit, which continuously removes $SF_6$ from the water for measurement using a membrane contactor, and an analytical unit composed of a gas chromatograph equipped with an electron capture detector (GC/ECD). The system has a detection limit of $1 \times 10^{-14}$ mol L$^{-1}$ and an analytical precision of ±1% (Ho et al., 2018a). A personal computer showed the $SF_6$ concentration in real time,
which guided the boat navigation and spatial survey.

Near the center of the $SF_6$ patch between 1 and 8 April 2015, we conducted 26 total stations for depth profiles measurements of temperature and salinity with a conductivity, temperature, and depth (CTD) sonde and we took discrete $^3$He and $SF_6$ samples at a subset of these stations (green triangles in Fig. 1). In total, we collected 16 $^3$He samples (~40 mL each) and 84 discrete $SF_6$ samples. $^3$He samples were taken in copper tubes mounted in aluminum channels and sealed at the ends with stainless steel
clamps. In the shore-based laboratory at the end of the experiment, $^3$He and other gases were extracted from the water in the copper tubes, transferred to flame-sealed glass ampoules, and measured using a He isotope mass spectrometer (Ludin et al., 1998). 84 discrete $SF_6$ samples were taken at the same stations using 50-mL glass syringes and submerged in water in a cooler until measurement back on shore at the end of each day. $SF_6$ was extracted by a headspace technique and measured on a GC/ECD as described by Wanninkhof et al. (1987). We used the mean $^3$He and $SF_6$ concentration for each day to determine $k$,
so there are six $^3$He/$SF_6$ data points between 3 and 8 April (Fig. 2f).

## 2.3 Measurements of environmental variables

We measured wind speed, wind direction, and air temperature at ~5 m above sea level every 10 s using a sonic anemometer (Vaisala WMT700) near Bob Allen Keys (25.02663°N, 80.68137°W; blue dot in Fig. 1). The air temperature was averaged
every 1 h to calculate the air-sea temperature difference (sea minus air). Additional wind speeds measured using a sonic

anemometer (Vaisala WXT532) at ~3 m above the sea level at 25.07209°N, 80.73511°W (pink square in Fig. 1, 7.4 km away from the blue dot) between 2015 and 2019 were obtained from U.S. National Park Service (NPS) (https://www.ndbc.noaa.gov/) to compare $k$ derived from this study and $k$ estimated from published parameterizations.

Wind speed data were extrapolated to 10 m above the sea level using the following equation (Amorocho and DeVries, 1980):

$$u_z = u_{10}\left(1 - C_{10}^{\frac{1}{2}}\kappa_c^{-1}\ln(10/z)\right) \tag{1}$$

where $u_z$ is the wind speed at height $z$, $\kappa_c$ is the von Kármán constant (0.41), $C_{10}$ is the surface drag coefficient of wind at 10 m height ($1.3\times10^{-3}$) (Stauffer, 1980).

Hourly tidal amplitude, sea surface temperature, and salinity data from the same site (blue dot in Fig. 1) between 2015 and 2019 were obtained from NPS (https://www.ndbc.noaa.gov/). The tidal amplitude was measured using a digital shaft encoder (WaterLog H331), and sea surface temperature and salinity were measured using multiparameter sondes (Hydrolab Quanta until 5 March 2019; OTT-Hydromet OTT-PLS-C thereafter).

We measured the $p\mathrm{CO}_2$ along the boat track (red lines in Fig. 1) using an underway system based on the design of Ho et al. (1997b) and incorporating the suggestions from Pierrot et al. (2009). Water was pumped through a thermosalinograph (TSG) into a showerhead equilibrator, and a high-precision thermistor measured the temperature. The gas was dried by Nafion and $\mathrm{Mg(ClO_4)_2}$ dryers, and was continuously circulated through a non-dispersive infrared (NDIR; LI-COR LI-840A) analyzer. We stopped the flow during measurement and vented the NDIR cell to the atmosphere. The interval between measurements was 41 s. Atmospheric air was taken from an inlet at the bow of the boat through a length of aluminum/plastic composite tubing (Dekabon), and was diverted into the NDIR analyzer at specific times (every ~72 min). We calibrated the analyzer at regular time intervals (~72 min) with a 511 ppm $\mathrm{CO}_2$ standard calibrated with a primary standard from NOAA/ESRL/GMD and a $\mathrm{CO}_2$-free reference gas (UHP $\mathrm{N}_2$ passed through soda lime to remove $\mathrm{CO}_2$). In total, 1,261 and 13 $x\mathrm{CO}_2$ data were taken from the water and air, respectively. With measured mole fraction of $\mathrm{CO}_2$ ($x\mathrm{CO}_2$), barometric pressure (P), and water vapor pressure at water surface temperature (Vp), we calculated the water and atmospheric $p\mathrm{CO}_2$ by applying the following expression (DOE, 1994): $p\mathrm{CO}_2 = (\mathrm{P}-\mathrm{Vp}) \times x\mathrm{CO}_2$. $p\mathrm{CO}_2$ values were corrected for temperature shifts in the sample from the intake point (i.e., as measured by the TSG) to the $p\mathrm{CO}_2$ system using an empirical equation proposed by Takahashi et al. (1993). Fugacity of $\mathrm{CO}_2$ ($f\mathrm{CO}_2$) was calculated by $f\mathrm{CO}_2 = \alpha \times p\mathrm{CO}_2$, where $\alpha$ is an activity coefficient calculated from a formula in Wanninkhof and Thoning (1993). Additional $f\mathrm{CO}_2$ data were obtained from National Oceanic and Atmospheric Administration (NOAA) Pacific Marine Environmental Laboratory (https://www.pmel.noaa.gov/) at 24.90°N, 80.62°W (cyan diamond in Fig. 1, 15 km away from the blue dot). $\mathrm{CO}_2$ flux between air and water was calculated with solubility ($\mathrm{K}_0$) and $f\mathrm{CO}_2$ using the equation below:

$$F = k\mathrm{K}_0(f\mathrm{CO}_{2water} - f\mathrm{CO}_{2air}), \tag{2}$$

where the $\mathrm{K}_0$ was calculated from the measured temperature and salinity (Weiss, 1974).

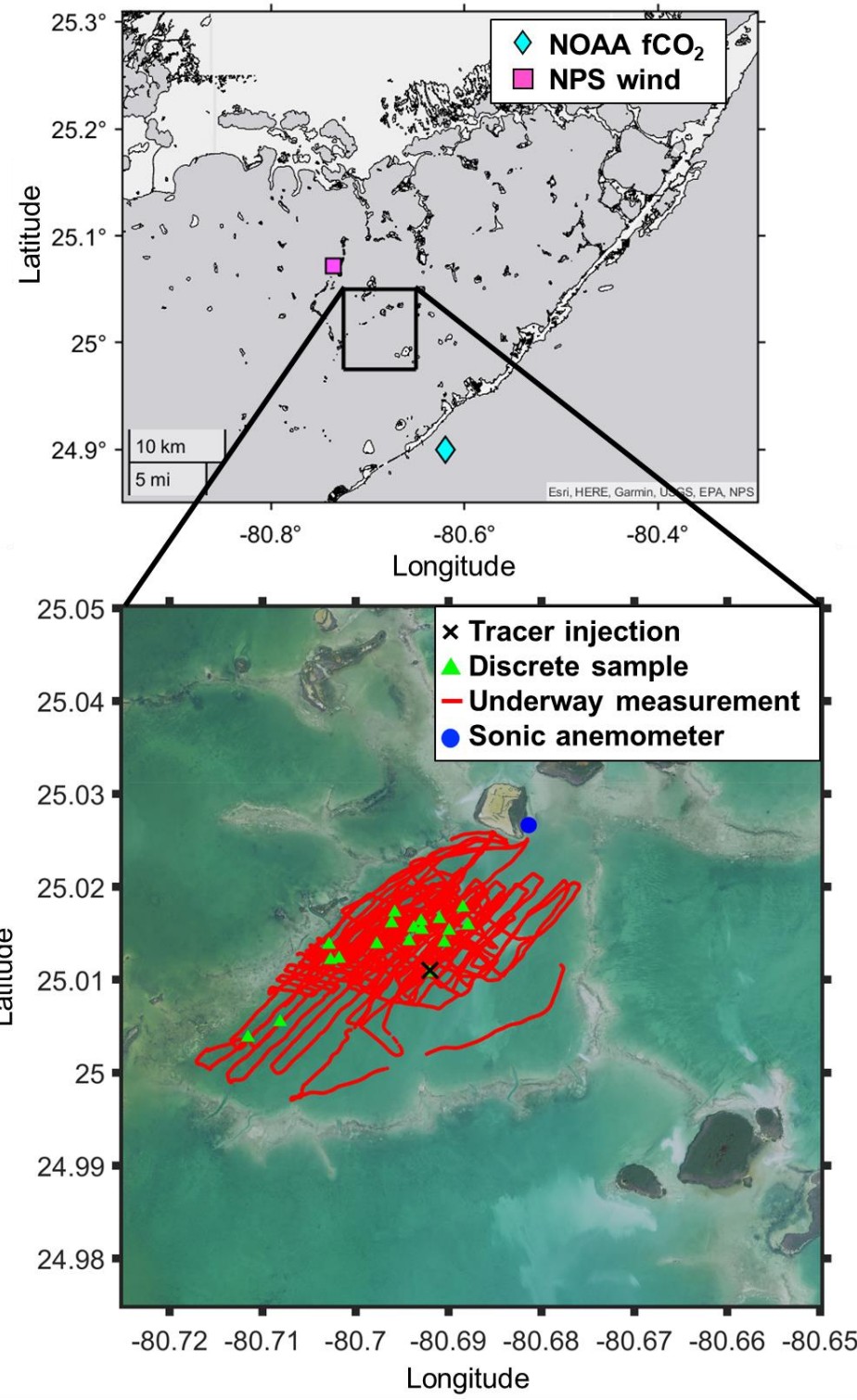

**Fig. 1** Map of the study area. The black cross is the $^3$He and $SF_6$ injection location; red lines are the boat track where underway measurement was conducted for $pCO_2$ and $SF_6$; the blue dot is the location where wind velocity, air temperature, water temperature, salinity, and tidal amplitude were measured; the green triangle is the stations where discrete samples for $^3$He and $SF_6$ were taken; the pink square is where additional wind velocity was measured; and cyan diamond indicates where additional $fCO_2$ were measured. Note that water temperature, salinity, tidal amplitude, and additional wind velocity were obtained from NPS, and additional $fCO_2$ data were obtained from NOAA Pacific Marine Environmental Laboratory. Map data are generated by MATLAB geobasemap "darkwater" and downloaded from the Fish and Wildlife Research Institute (https://myfwc.com/research/) and NOAA Office for Coastal Management (https://www.noaa.gov/)

2.4. Gas transfer velocity calculation and $^3$He/$SF_6$ ratio modeling

The $^3$He/$SF_6$ technique relies on the well-tested assumption that patch dilution, such as by horizontal mixing, affects the individual tracer concentrations but does not alter the $^3$He/$SF_6$ ratio; the only process that changes the $^3$He/$SF_6$ ratio is air-sea gas exchange. The gas transfer velocity for $^3$He, $k_{^3\text{He}}$, can be determined as follows (Wanninkhof et al., 1993):

$$k_{^3\text{He}} = -\left(1 - \left(Sc_{SF_6}/Sc_{^3\text{He}}\right)^{-1/2}\right)^{-1} h \frac{d}{dt}\left(\ln\left(^3\text{He}_{exc}/SF_6\right)\right) \tag{3}$$

where $Sc_{SF_6}$ and $Sc_{^3\text{He}}$ are the Schmidt numbers (i.e., the kinematic viscosity of water divided by diffusion coefficient of the gas in water) for $SF_6$ and $^3$He, respectively (see section 2.5). $h$ is the measured water depth in Florida Bay, adjusted for tidal variation. $^3$He$_{exc}$ is the $^3$He in excess of solubility equilibrium with the atmosphere (used interchangeably with $^3$He here). The gas transfer velocity measured during this experiment is normalized to $k(600)$, where 600 corresponds to $Sc$ number of $CO_2$ in freshwater at 20°C:

$$k(600) = k_{^3\text{He}}\left(600/Sc_{^3\text{He}}\right)^{-1/2}. \tag{4}$$

The decrease of the $^3$He/$SF_6$ ratio was compared to the decrease predicted by published wind speed/gas exchange parameterizations to assess the validity of these parameterizations for the study area. Under the assumption that air-sea gas exchange is the only process that alters the $^3$He/$SF_6$ ratio in the water, the change in $^3$He/$SF_6$ ratio during this experiment can be modeled by an analytical solution to equation (3):

$$\left(^3\text{He}/SF_6\right)_t = \left(^3\text{He}/SF_6\right)_{t-1} \exp\left(-\frac{k_{^3\text{He}}\Delta t}{h}\left(1 - \left(Sc_{SF_6}/Sc_{^3\text{He}}\right)^{-1/2}\right)\right) \tag{5}$$

where $\left(^3\text{He}/SF_6\right)_t$ is the $^3$He to $SF_6$ ratio at time $t$ and $\left(^3\text{He}/SF_6\right)_{t-1}$ is the ratio at the previous time step. $k_{^3\text{He}}$ is determined from wind speeds measured during the experiment and published parameterizations. The skill of these parameterizations to predict the measured $^3$He/$SF_6$ during this experiment is evaluated in terms of the coefficient of variation of the root mean square error (cvRMSE):

$$\text{cvRMSE} = \frac{\sqrt{\frac{1}{N}\sum_{n=1}^{N}\left(R_{mod}^n - R_{obs}^n\right)^2}}{\overline{R_{obs}}}, \tag{6}$$

where $R_{obs}^n$ and $R_{mod}^n$ are the observed and modeled $^3He/SF_6$ tracer ratios, respectively, and N is the number of stations sampled after the initial sampling (5 for table 2 and 2 for Fig. 5e). The ability of commonly used parameterizations, including the quadratic relationships of Wanninkhof (1992), Nightingale et al. (2000), and Ho et al. (2006), the exponential relationship of Raymond and Cole (2001), and the hybrid parameterization of Wanninkhof et al. (2009) to predict $k$ in Florida Bay was evaluated by examining the cvRMSE. Equation (6) was also used to derive the optimal coefficients (A) for a quadratic ($k = Au_{10}^2$) parameterization by minimizing the cvRMSE. We regarded A with minimum cvRMSE as the best coefficient for parameterization.

## 2.5 Calculation of $Sc$ number

In the literature, $Sc$ is often calculated from a compilation by Wanninkhof (2014). However, because the salinity in Florida Bay was higher than the range provided by Wanninkhof (2014), we re-calculated $Sc$ for an extended salinity range. The kinematic viscosities for freshwater and seawater were determined using equations given by Sharqawy et al. (2010), and the molecular diffusion coefficients of various gasses for freshwater were calculated using empirical equations derived from previous studies (Jähne et al., 1987; Wilke and Chang, 1955; Hayduk and Laudie, 1974; King and Saltzman, 1995; Saltzman et al., 1993; Zheng et al., 1998; De Bruyn and Saltzman, 1997). While the effect of temperature on molecular diffusion coefficient is well studied, the effect of salinity has been the subject of fewer investigations. For $SF_6$, $CH_3Br$, and CFC-11, the diffusion coefficients in seawater are similar to those in freshwater (King and Saltzman, 1995; De Bruyn and Saltzman, 1997; Zheng et al., 1998). However, the diffusion coefficients for methane ($CH_4$), CFC-12, and He in seawater are 4-7% lower than those in freshwater (Jähne et al., 1987; Saltzman et al., 1993; Zheng et al., 1998). To represent the dependence of molecular diffusion coefficients on salinity for gases other than $SF_6$, $CH_3Br$, and CFC-11, we linearly interpolated/extrapolated the molecular diffusion coefficients for different salinities by assuming that the diffusion coefficients decrease by 6% when the salinity is 35 compared to freshwater (Jähne et al., 1987; Wanninkhof, 2014). This assumption suggests that the diffusion coefficients for a salinity of 40 are approximately 7% smaller than those for freshwater. A least-squares fourth-order polynomial fit, incorporating the effect of salinity, was used to predict Sc values at various temperatures and salinities (Table 1).

Table 1. Coefficients for a least-squares fourth-order polynomial fit of Schmidt number versus salinity and temperature for various salinity and temperatures from 0 to 40°C.

| Gas | A | a | B | b | C | c | D | d | E | e | Sc number (20°C | Sc number (20°C |
|---|---|---|---|---|---|---|---|---|---|---|---|---|

| | | | | | | | | | |  |  |
|---|---|---|---|---|---|---|---|---|---|---|---|
| $^3$He | 334.38 | 0.90630 | -17.566 | -0.040902 | 0.53156 | 0.0011076 | -0.0094081 | $1.8342\times10^{-5}$ | $7.1715\times10^{-5}$ | $1.3483\times10^{-7}$ | 132 | 146 |
| He | 377.10 | 1.1097 | -19.810 | -0.050665 | 0.59949 | 0.0013852 | -0.010610 | $2.3081\times10^{-5}$ | $8.0880\times10^{-5}$ | $1.7028\times10^{-7}$ | 149 | 166 |
| Ne | 764.44 | 2.2495 | -43.818 | -0.11364 | 1.3943 | 0.0032933 | -0.025331 | $5.652\times10^{-5}$ | 0.00019561 | $4.2289\times10^{-7}$ | 274 | 306 |
| Ar | 1876 | 5.5663 | -131.69 | -0.32458 | 4.9298 | 0.010744 | -0.099518 | 0.00020223 | 0.00081784 | $1.5998\times10^{-6}$ | 549 | 619 |
| $O_2$ | 1733.6 | 5.1437 | -121.69 | -0.29994 | 4.5556 | 0.0099283 | -0.091963 | 0.00018688 | 0.00075576 | $1.4784\times10^{-6}$ | 507 | 572 |
| $N_2$ | 2080.6 | 6.1735 | -146.06 | -0.35999 | 5.4677 | 0.011916 | -0.110337 | 0.00022429 | 0.00090706 | $1.7743\times10^{-6}$ | 609 | 687 |
| Kr | 2036.2 | 5.9923 | -133.13 | -0.35181 | 4.5886 | 0.011183 | -0.087051 | 0.00020169 | 0.00068746 | $1.5474\times10^{-6}$ | 623 | 695 |
| Xe | 2688.8 | 7.9128 | -181.43 | -0.48144 | 6.3779 | 0.015655 | -0.122233 | 0.00028594 | 0.00091717 | $2.2082\times10^{-6}$ | 788 | 880 |
| $CH_4$ | 1900.3 | 5.5923 | -119.02 | -0.31267 | 3.9947 | 0.0096399 | -0.074686 | 0.00017095 | 0.00058531 | $1.3002\times10^{-6}$ | 614 | 685 |
| $CO_2$ | 1914.2 | 5.6330 | -123.18 | -0.32481 | 4.2040 | 0.0102008 | -0.079322 | 0.00018296 | 0.00062463 | $1.3992\times10^{-6}$ | 598 | 667 |

| | A | a | B | b | C | c | D | d | E | e | | |
|---|---|---|---|---|---|---|---|---|---|---|---|---|
| N$_2$O | 2127 | 6.3112 | -149.31 | -0.36802 | 5.5897 | 0.012182 | -0.11284 | -0.00022929 | 0.0009273 | 1.8139×10$^{-6}$ | 622 | 702 |
| Rn | 3154.1 | 9.2820 | -220.51 | -0.58779 | 7.9274 | 0.019612 | -0.15397 | -0.00036344 | 0.0012313 | 2.8283×10$^{-6}$ | 880 | 982 |
| SF$_6$ | 3024 | 3.0926 | -193.63 | -0.14258 | 6.5878 | 0.0035655 | -0.12409 | -5.3058×10$^{-5}$ | 0.00097626 | 3.5673×10$^{-7}$ | 950 | 996 |
| DMS | 2582.0 | 7.5983 | -160.71 | -0.42182 | 5.3733 | 0.012946 | -0.10025 | -0.00022905 | 0.00078480 | 1.7401×10$^{-6}$ | 841 | 938 |
| CFC-12 | 3460.3 | 10.183 | -225.72 | -0.5963 | 7.7688 | 0.018924 | -0.14727 | -0.00034099 | 0.0011625 | 2.6148×10$^{-6}$ | 1061 | 1184 |
| CFC-11 | 3446.9 | 3.5251 | -214.51 | -0.15589 | 7.1717 | 0.0037741 | -0.1338 | -5.4896×10$^{-5}$ | 0.0010474 | 3.6378×10$^{-7}$ | 1123 | 1176 |
| CH$_3$Br | 2101 | 2.1487 | -133.27 | -0.097717 | 4.5081 | 0.0024177 | -0.084644 | -3.571×10$^{-5}$ | 0.00066483 | 2.3896×10$^{-7}$ | 668 | 700 |
| CCl$_4$ | 3973.3 | 11.789 | -278.92 | -0.68747 | 10.442 | 0.022756 | -0.21078 | -0.00042832 | 0.0017322 | 3.3884×10$^{-6}$ | 1163 | 1312 |

$Sc = A+aS + (B+bS)T + (C+cS)T^2 + (D+dS)T^3 + (E+eS)T^4$ (T in °C). The last two columns are the calculated Schmidt number for 20°C, and salinities of 0 and 35 as examples, respectively. The diffusion coefficients, denominators of $Sc$, are derived from the following: $^3$He, He, Ne, Kr, Xe, CH$_4$, CO$_2$ and Rn measured by Jähne et al. (1987); Ar, O$_2$, N$_2$, N$_2$O, and CCl$_4$ fit from Wilke and Chang (1955) adapted by Hayduk and Laudie (1974); SF$_6$ measured by King and Saltzman (1995); DMS measured by Saltzman et al. (1993); CFC-11 and CFC-12 measured by Zheng et al. (1998); CH$_3$Br measured by De Bruyn and Saltzman (1997). $Sc$ numbers for temperature of 20°C and salinity of 35 become larger than $Sc$ numbers for temperature of 20°C and salinity of 0 by 4.7–4.8% for SF$_6$, CFC-11 and CH$_3$Br and 10.8–12.8% for other gasses, respectively. Note that the fits are based on simple assumptions (see section 2.5), and the dependence of $Sc$ on salinity needs to be investigated further in the future.

## 3. Results and discussion

3.1 Environmental parameters

During the experiment, wind was predominately from the east, and wind speeds increased towards the latter part of the study period (Fig. 2a). The mean and the standard deviation of the wind speed during the study period was $5.5 \pm 2.0$ m s$^{-1}$ (range=0.12–12 m s$^{-1}$). Water temperature showed a diurnal pattern with a mean and standard deviation of $26.3 \pm 1.3°C$ (Fig. 2b). The diurnal pattern of the air temperature was weak, with a mean and standard deviation of $25.1 \pm 0.6°C$. The air-sea temperature difference showed diurnal cycles, which were mainly driven by the diurnal cycle of the sea temperature, consistent with observations by Van Dam et al. (2020). Salinity remained consistent throughout the study period ($41 \pm 0.1$; not shown). The tide consistently showed semidiurnal cycles with an amplitude of $\leq 0.2$ m throughout the study period.

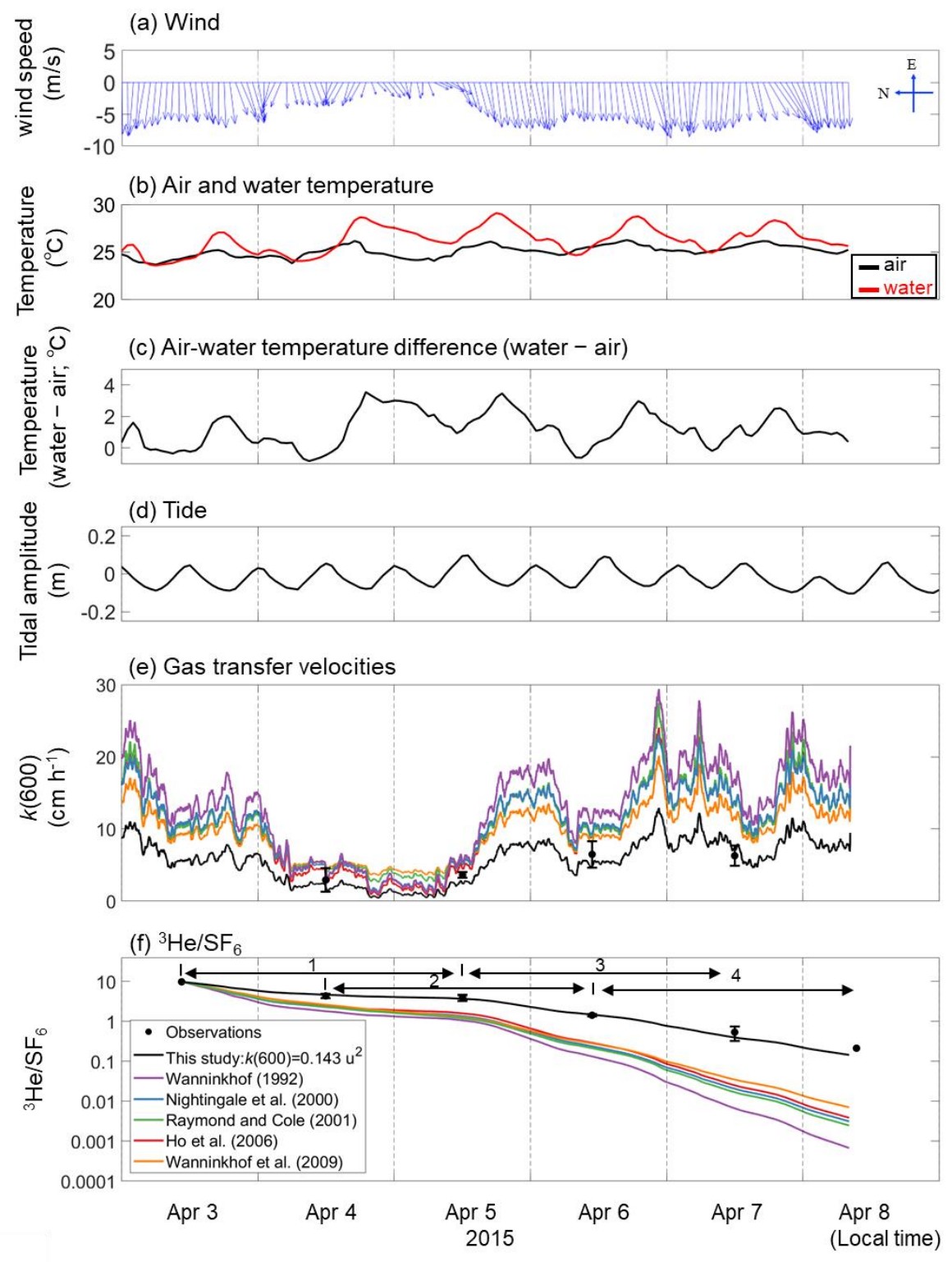

**Fig. 2** Time series of (a) hourly averaged wind vector at 10 m height (m s$^{-1}$), (b) water temperature and air temperature (°C), (c) temperature difference (water temperature minus air temperature; units: °C), (d) tidal amplitude (units: m) and (e) measured and estimated $k(600)$ which is the gas transfer velocities normalized for 20°C freshwater $CO_2$ and (f) measured and modeled change in $^3$He/SF$_6$. Note that the wind direction is towards the north when the vector is towards the left. The time zone is local time. The figure legend for (e) is the same as that in (f). The numbers in (f) indicate the periods corresponding to the x-axis in Fig. 5.

3.2 Gas transfer velocity in Florida Bay and assessment of published parameterization

The measured $k(600)$ was 4.8 ± 1.8 cm h$^{-1}$ (mean ± s.d.) (Figs. 2e and 3), which was lower than previous studies conducted in coastal and open oceans at the same wind speed (Fig. 3 of Ho & Wanninkhof, 2016). A new parameterization was produced based on results from this experiment by minimizing the cvRMSE of $A \cdot u_{10}^2$, where A is a coefficient (Fig. 4):

$$k(600) = 0.143u_{10}^2 \tag{7}$$

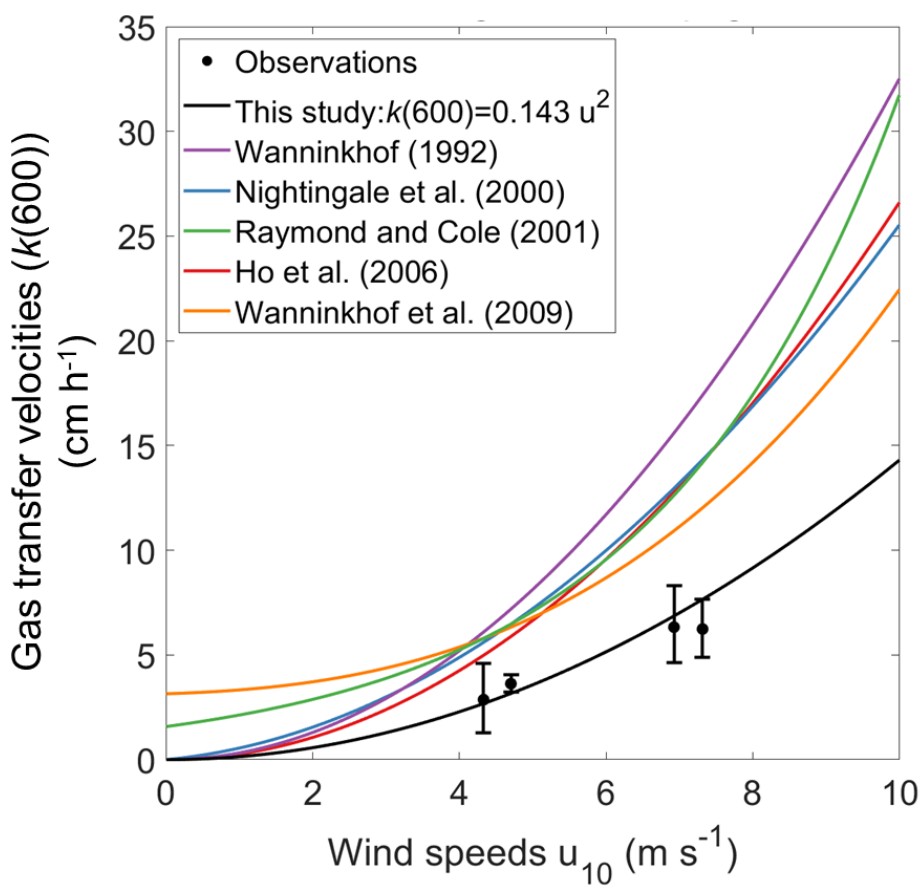


**Fig. 3** Measured and modeled $k(600)$ (units: cm h$^{-1}$) with wind speed at 10 m height (units: m s$^{-1}$).

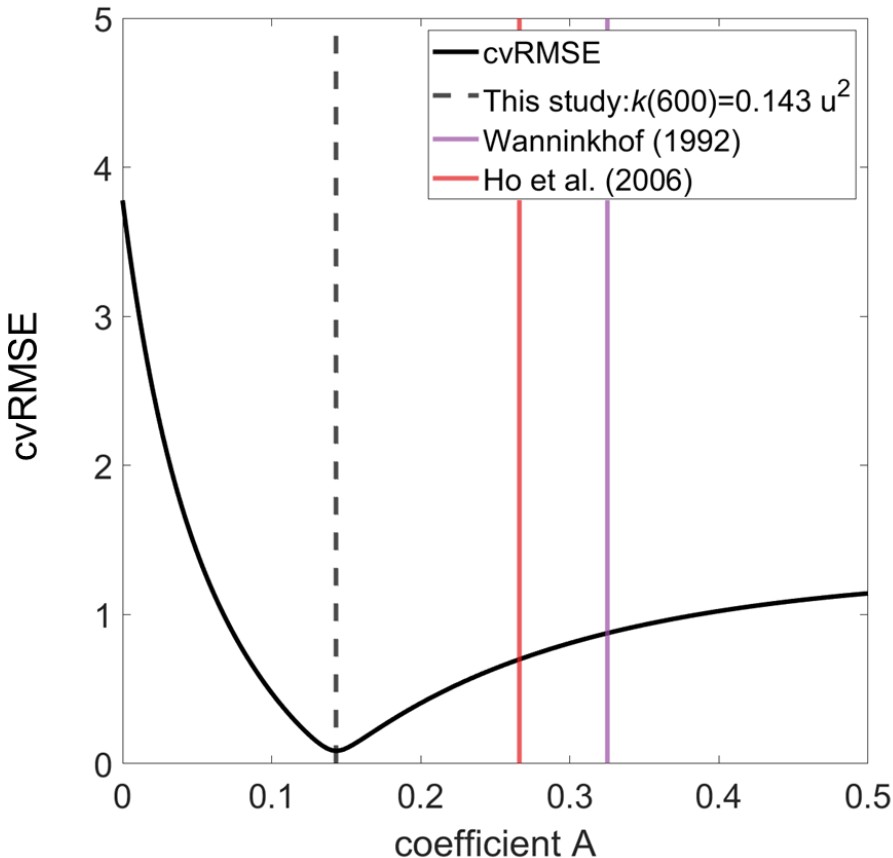

**Fig. 4** The relationship between cvRMSE and the coefficient A in the equation $k(600)=A\ u_{10}^2$. Three vertical lines indicate the coefficients derived from this study, as well as those of Ho et al. (2006) and Wanninkhof (1992) from left to right. Note that $k(660)$ in Wanninkhof (1992) was converted to $k(600)$ by assuming that they scale as $Sc$ to the power of $-1/2$.

The cvRMSE between the measured $^3$He/SF$_6$ and this new parameterization, equation (7), was 8.6%, while the cvRMSEs calculated from previously published wind speed/gas exchange parameterizations were more than 70% (Table 2). The coefficient of 0.143 was 46% and 56% lower than the $k(600)$ of 0.266 and 0.325 from Ho et al. (2006) and Wanninkhof (1992), respectively (Fig. 3). The result of previous studies that used the parameterization of Wanninkhof (1992) in Florida Bay was modified in section 3.3. The estimated $k(600)$ derived from equation (7) was 5.5±3.0 cm h$^{-1}$, while all the published parameterizations estimated over 9.0 cm h$^{-1}$ on average between 3 and 8 April 2015 (Table 2 and Fig. 2e). $k$ for CO$_2$ at in-situ temperature and salinity between 2015 and 2019 were also calculated using equation (7) and the previously published

parameterizations (Table 3). Annual averaged $k$ ranged between 3.7–4.3 cm h$^{-1}$ in Florida Bay between 2015 and 2019, while published parameterization would yield values of 6.9–11.6 cm h$^{-1}$.

Table 2. Gas transfer velocities determined from this study and published parameterization.

| References | Parameterization | Mean $k(600)$ (cm h$^{-1}$) | cvRMSE |
|---|---|---|---|
| This study | $k(600)= 0.143u_{10}^2$ | 5.5±3.0 | 8.6% |
| Wanninkhof (1992) | $k(660)=0.31u_{10}^2$ | 12.4±6.8 | 87.5% |
| Nightingale et al. (2000) | $k(600)=0.333u_{10} + 0.222u_{10}^2$ | 10.4±5.2 | 76.0% |
| Raymond and Cole (2001) | $k(600)=1.58e^{0.3u_{10}}$ | 10.7±5.3 | 78.2% |
| Ho et al. (2006) | $k(600)=0.266u_{10}^2$ | 10.2±5.5 | 70.0% |
| Wanninkhof et al. (2009) | $k(660)=3+0.1u_{10} + 0.064u_{10}^2 + 0.011u_{10}^3$ | 9.4±3.8 | 73.1% |

The observed $k(600)$ was $4.8 \pm 1.8$ cm h$^{-1}$ (average ± standard deviation). Note that $k(660)$ was converted to $k(600)$ by assuming
that the scale by $Sc$ to the power of $-1/2$.

Table 3. Gas transfer velocities of $CO_2$ at in-situ temperature and salinity in Florida Bay between 2015 and 2019 calculated using the wind speed/gas exchange parameterization determined here and from published parameterizations.

| Parameters | | 2015 | 2016 | 2017 | 2018 | 2019 | Between 2015 and 2019 |
|---|---|---|---|---|---|---|---|
| Wind speed (m s$^{-1}$) | | 4.2±2.4 | 4.4±2.6 | 4.1±2.7 | 4.4±2.5 | 4.7±2.4 | 4.4±2.5 (range=0–27.5) |
| Sea temperature (°C) | | 27.6±3.8 | 26.7±4.1 | 26.8±3.9 | 26.7±4.2 | 27.3±3.8 | 27.0±4.0 (range=13.2–36.2) |
| Salinity | | 40.5±4.0 | 34.1±3.6 | 34.7±5.2 | 33.0±2.7 | 39.5±2.9 | 36.3±4.8 (range=24.5–51.8) |
| Tidal amplitude (m) | | 0.16±0.03 | 0.17±0.03 | 0.18±0.06 | 0.18±0.03 | 0.18±0.03 | 0.17±0.04 (range=0.04–0.33) |
| Mean $k$ for $CO_2$ (cm h$^{-1}$) | This study | 3.7±4.0 | 4.0±4.1 | 3.8±5.6 | 4.0±4.1 | 4.3±4.4 | 3.6±4.2 (range=0–108) |
| | Wanninkhof (1992) | 8.4±9.1 | 9.0±9.4 | 8.7±12.8 | 9.1±9.4 | 9.9±9.9 | 8.3±9.6 (range=0–247) |
| | Nightingale et al. (2000) | 7.3±7.0 | 7.8±7.3 | 7.4±9.6 | 7.8±7.3 | 8.5±7.6 | 7.1±7.4 (range=0–176) |
| | Raymond and Cole (2001) | 8.5±8.9 | 8.9±9.1 | 11.6±86.2 | 9.1±10.4 | 9.7±10.0 | 8.7±36 (range=1.6–6127) |
| | Ho et al. (2006) | 6.9±7.4 | 7.4±7.7 | 7.1±10.4 | 7.4±7.7 | 8.1±8.1 | 6.8±7.9 (range=0–202) |
| | Wanninkhof et al. (2009) | 7.8±5.4 | 8.1±5.5 | 8.2±10.1 | 8.1±5.7 | 8.6±6.0 | 7.4±6.4 (range=3.1–298) |

The standard deviation of Raymond and Cole (2001) was large in 2017 since wind speed was as high as 27.5 m s$^{-1}$, and $k$ was
as high as $6.1 \times 10^3$ cm h$^{-1}$.

The deviations between observed and modeled $^3$He/SF$_6$, which is derived from published parameterizations, become larger with time (Figure 2f). This indicates that the published parameterizations overpredict $k$ in Florida Bay, which is consistent with the findings of Van Dam et al. (2020).

Van Dam et al. (2020) estimated $k$ using heat as a proxy ($k_H$) in Florida Bay. They found that $k_H$ was lower than $k$ calculated from published parameterization even though $k_H$ is known to overpredict $k$. They suggested that the stratification due to temperature restricts air-sea gas exchange since the deviation between $k_H$ and $k$ from commonly-used parameterization was large when the air-sea temperature difference was large. To investigate the relationship between environmental parameters and the deviation between measured and estimated air-sea gas exchange, we examined the relationship between temperature

difference and the deviation between observation and the models by calculating cvRMSE separately in four periods (Fig. 5). We found no clear relationship between the deviation and air-sea temperature difference. The deviation observed in Van Dam et al. (2020) might be due to the fact that $k_H$ contains the air-sea temperature difference in its equation (equation 7 in Van Dam et al. 2020); $k_H$ becomes smaller when the air-sea temperature difference is large and vice versa. The deviation between observation and models was generally larger when wind speed was higher. cvRMSE became the largest values for all

parameterizations in Period 4 when the mean wind speed was 7.3 m s$^{-1}$.

The new wind speed/gas exchange parameterization predicts the observed change in $^3$He/SF$_6$ well (Fig. 2f and Table 2), suggesting that wind is the dominant factor controlling gas exchange in this area. In Florida bay, waves are damped by seagrasses (Prager and Halley, 1999), which could be one of the causes of lower $k$ in this study. There is also the possibility that limited wind fetch in this region led to relatively weak waves and turbulence compared to other regions, contributing to

lower $k$. Wind fetch is limited in this region since the wind mostly blows from east to west, and the Florida Keys restricts the water exchange between the bay and the Atlantic Ocean (Fig. 1 and Fig. 2a). There was almost no rainfall to affect $k$ during the study period. The tidal amplitude was small (~0.1 m) (Fig. 2d), suggesting that the bottom-generated turbulence was weak.


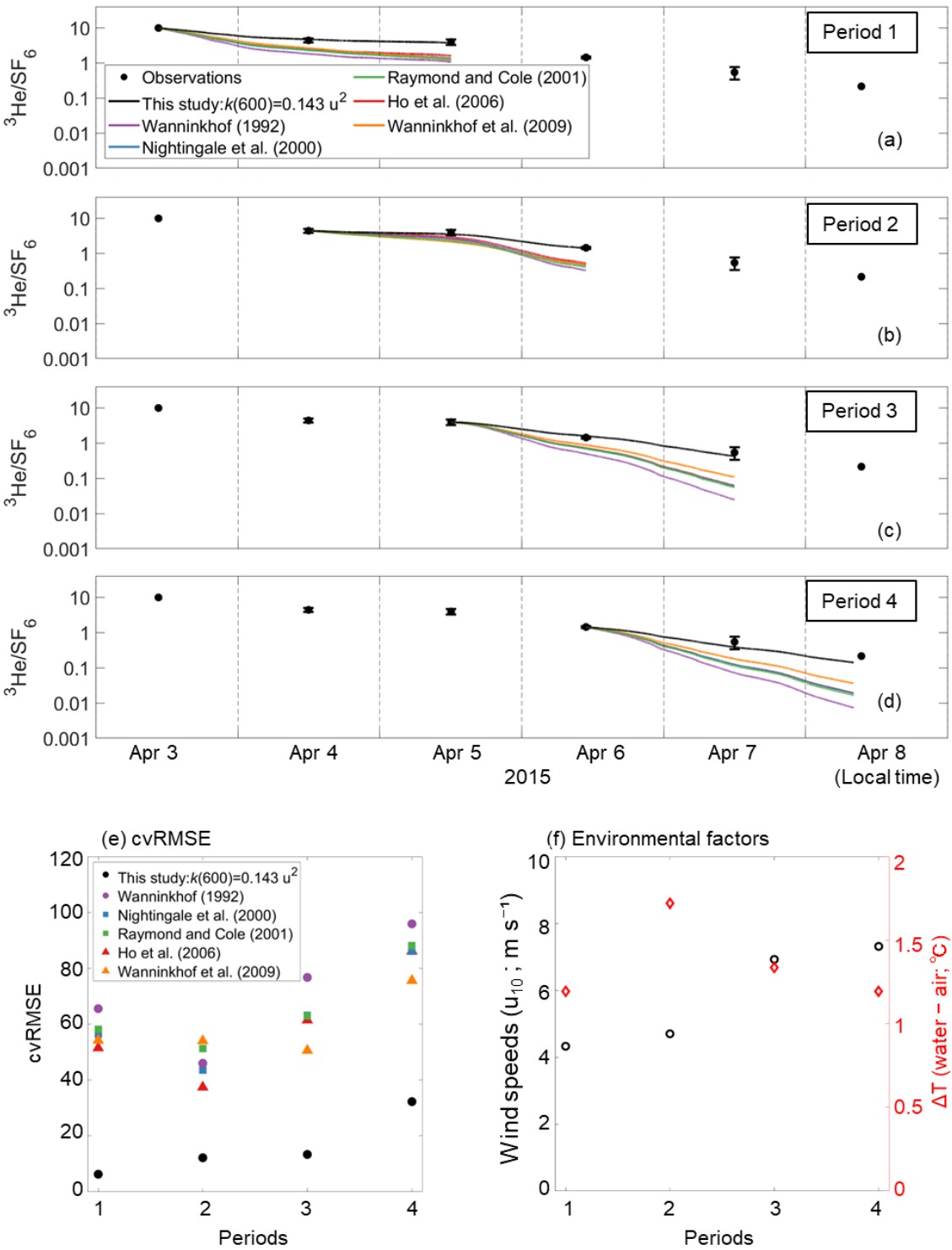

**Fig. 5** Time series of measured and modeled change in $^3$He/SF$_6$ in (a) period 1, (b) period 2, (c) period 3, and (d) period 4 in Fig. 2f. $^3$He/SF$_6$ value is set to the starting point of each period. (e) The cvRMSE, (f) mean wind speed (m s$^{-1}$) and air-sea temperature difference (°C) during the period of 1–4. The x-axis represents the periods in Fig. 2f.


3.3 Implications for biogeochemistry

Although the experiment was conducted over a short period of 8 days, our new parameterization, equation (7), fit the observations well; This implies that equation (7) can be applied even in different seasons and years if the wind speed is in the range of 0.12–12 m s$^{-1}$ and seagrass conditions are similar (dominant seagrass of *Thalassia testudinum* has 63.6 (range=0–

215) g dry weight m$^{-2}$ standing crop in Florida Bay (Zieman et al., 1989)). The parameterization determined in this study should be applicable to other seagrass ecosystems as well since seagrass ecosystems are typically in coastal regions. In these environments, waves are damped by seagrasses and limited fetch. This wind speed/gas exchange parameterization proposed here might be applicable not only in seagrass ecosystems but also in other wind-fetch limited areas. To assess the applicability of this new parameterization in other inland ecosystems, additional $^3$He/SF$_6$ dual tracer experiments will need to be conducted.

Specifically, measuring the seagrass density and conducting dual-tracer experiments simultaneously is needed to relate the $k$ and vegetation distribution.

The observed daytime $p$CO$_{2water}$ and $p$CO$_{2air}$ were 224 ± 12 and 391 ± 3 µatm, respectively (Fig. 6a). The $p$CO$_{2water}$ of 224 ± 12 µatm was in the range shown by Zhang and Fischer (2014), who examined the $p$CO$_{2water}$ in the whole basin of the Florida Bay from 2006 to 2012, and showed that $p$CO$_{2water}$ minimum was ~200 µatm in April (Fig. 3 of Zhang and Fischer 2014).

Since the observed $p$CO$_{2water}$ was lower than $p$CO$_{2air}$, CO$_2$ goes from the air to the sea during the daytime in the observation period (between 3 and 8 April 2015). The calculated daytime CO$_2$ flux using the measured $p$CO$_2$ difference and modeled $k$ in this study (Black solid line in Fig. 2e) was –5.3 ± 3.0 mmol m$^{-2}$ day$^{-1}$ (negative value denotes CO$_2$ flux from the air to the water) (Fig. 6b). The CO$_2$ flux varied both within a day and between days mainly due to the variability in $k$ (Note that $k$(600) in Fig. 2e is filtered with 25 minutes running average). Diurnal fCO$_{2water}$ amplitude at the NOAA station (cyan diamond in Fig.

1) between 3 and 8 April 2015 was as small as 25–53 µatm, and so we calculated daily CO$_2$ flux by assuming CO$_2$ difference between air and water during the night is the mean daytime CO$_2$ difference. The calculated daily CO$_2$ flux was –7.0 ± 3.5 mmol m$^{-2}$ day$^{-1}$, which was higher than daytime CO$_2$ flux because wind speed was higher during the night.

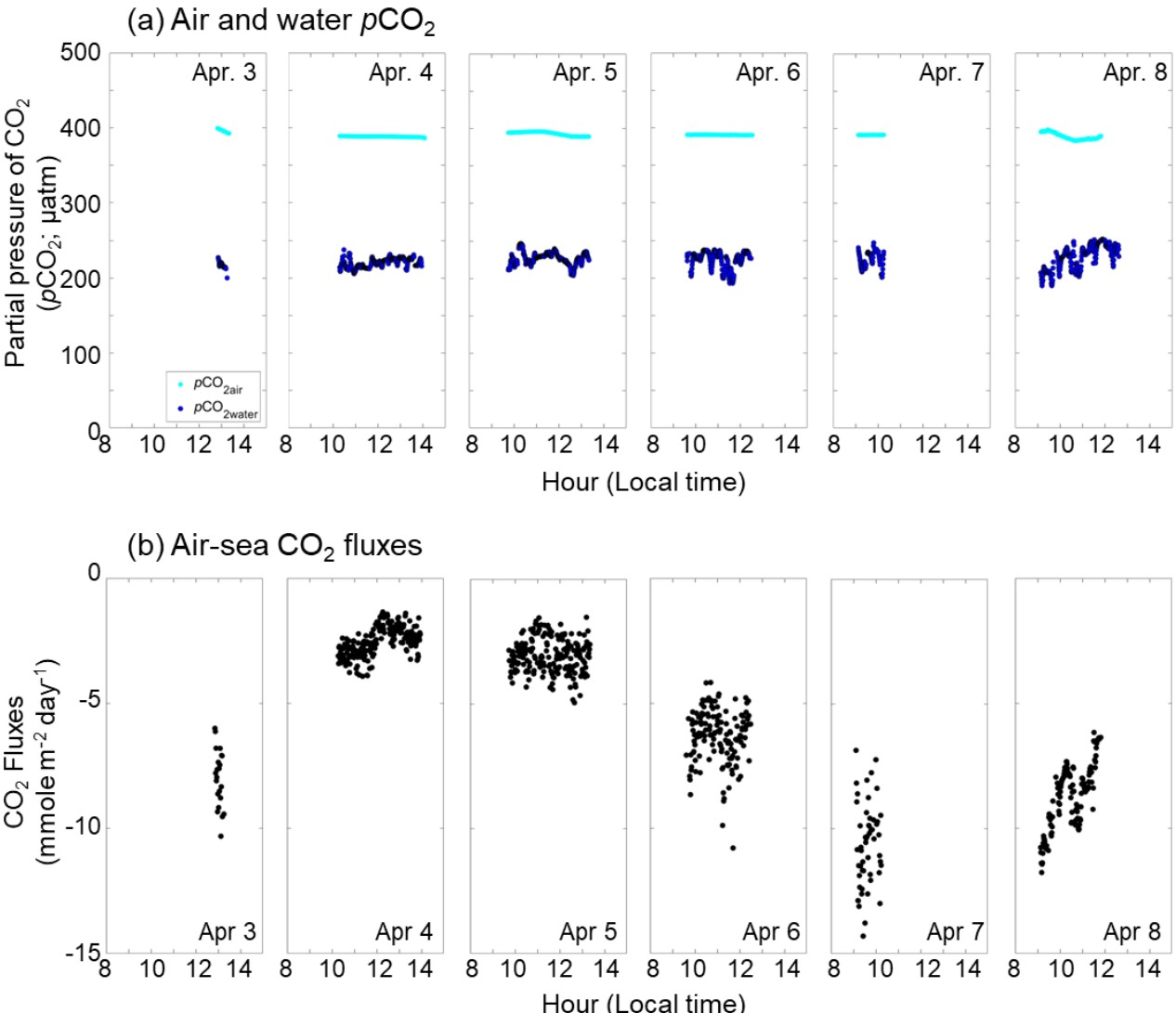

**Fig. 6** Time series of (a) measured $pCO_{2water}$ (blue dots) and $pCO_{2air}$ (cyan dots) (units: µatm), (b) calculated $CO_2$ flux (units: mmole $m^{-2}$ $day^{-1}$). Negative $CO_2$ flux indicates that the sea is a sink of $CO_2$. The time zone is local time.

Contrary to our experimental period, the annual averaged $CO_2$ flux is known to be from the water to the air in Florida Bay (e.g., Zhang and Fischer, 2014; Van dam et al., 2021). The $pCO_2$ and $CO_2$ flux in Florida Bay are suggested to have seasonality due to cyanobacteria blooms (Zhang and Fischer, 2014). The seasonality of seagrasses may also contribute to the seasonality of $pCO_2$ and $CO_2$ flux, as its productivity also shows seasonality (higher in spring and summer and lower in fall and winter) (Zieman et al., 1999). Zhang and Fischer (2014) measured the $pCO_{2water}$ for the whole area of Florida Bay and estimated the $CO_2$ flux in Florida Bay to be $3.93 \pm 0.91$ mol $m^{-2}$ $yr^{-1}$ using the parameterization of Wanninkhof (1992); we recalculated the

$CO_2$ flux to be $1.73 \pm 0.40$ mol m$^{-2}$ yr$^{-1}$ by multiplying 0.44 (e.g., 1 minus 0.56; see section 3.2). By conducting atmospheric
eddy covariance measurements near the Bob Allen Keys (blue dot in Fig. 1), Van Dam et al. (2021) showed that the $CO_2$ flux
in Florida Bay was 6.1–7.0 mol m$^{-2}$ yr$^{-1}$, which is significantly higher than the corrected value of $1.73 \pm 0.40$ mol m$^{-2}$ yr$^{-1}$.
Although the reason is not clear, primary production by phytoplankton and seagrasses might be lower when Van Dam
et al. (2021) conducted their observation (2019–2020), resulting in higher $CO_2$ flux from sea to air, since there is no negative
mean $CO_2$ flux in spring when they conducted their measurements (Fig. 1a in Van Dam et al., 2021). Van Dam et al. (2021)
also calculated the excess $CO_2$, which is the $CO_2$ concentration difference between water and air to achieve the annual $CO_2$
flux of 6.1–7.0 mol m$^{-2}$ year$^{-1}$, in Florida Bay to be between 5.2 and 6.0 μmol kg$^{-1}$, using a mean $k$ of 11.7 cm h$^{-1}$; we
recalculated the excess $CO_2$ to be between 14 and 16 μmol kg$^{-1}$ using the $k$ of 4.3 cm h$^{-1}$ parameterized from the current study
(Table 3). The recalculated excess $CO_2$ almost doubles the calculation of Van Dam et al. (2021) and requires significantly
more $CO_2$ input.


## 4. Summary

Air-sea gas exchange was investigated in a seagrass ecosystem in South Florida, USA, using the $^{3}$He and SF$_6$ dual tracer
technique. The gas transfer velocity was lower than that in other coastal areas and open oceans, and commonly-used wind
speed/gas exchange parameterizations overpredict the gas transfer velocities, especially when wind speeds were relatively high
($> 7$ m s$^{-1}$). A new wind speed/gas exchange parameterization was proposed ($k(600) = 0.143u_{10}^2$), which was able to predict
the observed gas transfer velocities significantly better than existing parameterizations. This result suggests that wind is the
dominant factor controlling gas exchange in the studied seagrass ecosystem, but the lower gas transfer velocity at a given wind
speed was due to limited wind fetch in the study area and wave attenuation by seagrass. To assess the wider applicability of
the proposed wind speed/gas exchange parameterization, more tracer release experiments are needed at similar inland
ecosystems.

**Data availability**

The data used for this article is found at https://doi.org/10.5281/zenodo.6730934. Click "Version Florida
10.5281/zenodo.7087773" in the right column.

**Author contributions**

DH conceived, designed, and conducted the experiment. RD performed the data analysis.

**Competing interests**

The authors have declared that they have no competing interests.

**Disclaimer**

**Acknowledgments**

We thank Nicholas Chow, Nathalie Coffineau, Benjamin Hickman, and Lindsey Visser for assistance in the field, Peter
Schlosser for measuring the $^3$He samples, Rik Wanninkhof for guidance on the Schmidt number calculations, Damon Rondeau at NPS for providing data on wind, temperature, salinity, and tide, Pierre Polsenaere and an anonymous reviewer for helpful comments.

**Financial support**

Funding was provided by the National Aeronautics and Space Administration (NNX14AJ92G) under the Carbon Cycle Science Program.

**Review statement**

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
