# Peer review of "Air-sea gas exchange in a seagrass ecosystem"

_EGUsphere, 2022_

## Referee Comment (RC2)

GENERAL COMMENTS:

The submitted manuscript of Dobashi and T. Ho under review for journal EGUsphere presents a gas transfer velocity determination using the dual tracer technique during a six days spring experiment over a seagrass ecosystem in south Florida. Controlling factors are examined and comparisons, conclusions and implications are drawn on gas transfer velocity parametrizations and air-sea $CO_2$ flux estimations over seagrass ecosystems.

The present study shows interesting analysis and computations on gas transfer velocity parameterizations over a seagrass meadow that will definitely give insights for the scientific community working and coastal carbon fluxes and processes. Nevertheless, as it stands, the submitted manuscript suffers from a lack of information (M&M section with in situ measurements and computations) and organization (especially the result and discussion section) which make difficult to follow and understand the approach and associated results (data set link is not mentioned in the MS sections, only at the end, maybe it can help but the link doesn't work). For instance, seagrass distribution/dynamic and other (than wind, temperature gradient) abiotic environmental parameters should be better described and linked to found gas transfer velocity parametrization. Also, some conclusions should be dampened a bit or reformulated particularly when authors compared their K600 parameterization obtained from a six days dataset in one season, to other different coastal ecosystems cited in the literature.

In this way, please see the specific comments below to help in the revision of the different sections and the overall manuscript. I would recommend further revisions to allow the publication of the present paper of Dobashi and T. Ho for the journal EGUsphere.

SPECIFIC COMMENTS:

Title:

It should be specified since very general in the present form.

Abstract:

- l.11-12: the general comparison between the present study and other coastal/open oceans is delicate since other environmental factors (than wind speeds) can be involved and controlled k.

- l.13-14 and 16-17: I would dampen this conclusion due to the limited data set and specific studied meadow compared to other seagrass systems worldwide (temperate, subtidal versus intertidal, depth, current, turbidity, rainfall, heat fluxes….).

- l.14: other settings? Please specify.

1 Introduction:

- l.21: an important part of seagrass above biomass (refractory matter) is also exported and does not sink to the bottom.

- l.22: Please update the reference adding other more recent works on carbon seagrass storage.

- l.25: high methane emissions from seagrass meadow have also been recently shown (see Schorn et al. 2021, PNAS 2022 Vol. 119 No. 9 e2106628119).

- l.26-27: the link with seagrass here is not clear, please specify.

- l.33: go further citing other involved parameters (and associated studies) in turbulence control. Also, it lacks in the introduction a review on existing works on gas transfer velocity determination according to (i) coastal ecosystem typology (lagoon, coastal ocean, estuary, seagrass presence or absence, ….) and (ii) methodology, for instance k can also be determined through simultaneous sea-air $CO_2$ fluxes using floating chamber, Eddy Covariance techniques and spatial water $pCO_2$, before focusing on the Florida Bay as done in the next paragraph.

- l.37: please be more specific citing for these studies the used methodology (k, atmospheric $CO_2$ exchanges) among wind speed/gas parametrizations and ecosystem typology as well (presence of seagrasses?).

2 Material and methods:

- l.49: could authors give an idea of seagrass densities in the 2000 km2 and in the specific studied area (Fig. 1). As it stands there is not enough information on spatial seagrass distribution, phenology (carbon stocks and fluxes, …) and concrete relationships done between seagrasses and k parametrization.

- l.60-65: figure caption should be specified and better linked do M&M following sections; it is in between in the submitted MS with cited measured parameters without information on how (sensors, frequencies, …) and when (duration) it was measured. The second zoomed figure should be even more restricted to better see the sampled area. In consequence, we are a bit lost when results and discussion section comes, i.e. parameter origins (sensor used, where, when, frequencies, duration, …).

- 2.2.: when and how long were tracers injected? Why?

- 2.3.: How many? Where (specify it in Fig. 1)? When?

- 2.4., l.89-90: where? When? How? L.92: black square in Fig. 1, how far form the site and why did authors use additional wind data? L.96-97: a given range for Z0 is given without any explanations on which ecosystem typology in the Florida Bay and how it was measured, please specify. Is the used average Z0 value enough precise for calculations? Maybe I did miss something but why Amorocho and DeVries (1980) equation was not used to compute wind speed data at 10 m?

- 2.5., l.100: $pCO_2$ measurements, used frequency? Where? L.106: "specific times", "at regular time intervals" could you specify it? L.107: $CO_2$ standard concentration?

- 2.7., l.158-159: the last sentence is a little bit awkward, in section 2.7 important testing was done to fit in a precise way coefficient Schmidt numbers especially according to temperature,

so why was it not done for salinity in a same way? L150-151: various salinity, you mean 0 and 35, don't you? Various temperatures from 0 to 40°C, could you specify (step, number of values?)? By the way, what are salinity range, mean values in Florida Bay in general and at your sampling area during the study? No result on it in the submitted MS?

3 Results and discussion:

- 3.1, l.183: how many k600 values were taken into account to obtain this average value? That information is lacking in the M&M section and in turn this result value is unclear and one might wonder if this result is comparable to other k600 values found in the bibliography (Fig. 2).

- Fig. 2: five stations were sampled according to M&M l.173. Four observations appear in Fig. 2, could author explain (is it 4 or 5 or anything else)?

- l.200-201: what did authors mean by "this parametrization" and "K600 between 2015 and 2019"? It is not clear. Could one have further elements or descriptions (sites, measurements, …) at minima instead of only having the two above references?

- l.202-203: and using old parametrizations, could we also have those values or at least element of comparisons (%, …)?

- l.205: I think Results and discussion section should be reorganized, presenting and fully describing first (which has not been done yet) temporal series of measured environmental parameters (grouping Figs. 4 and 6 or at least water pCO2 measurements for instance, Fig. 5) and then K600 descriptions and comparisons with plots (Figs. 2, 3) and tables and relevant controlling environmental factors on k600. As it stands, it is not possible for the reader to see how the ecosystem functioned during these six days experiments before understanding K600 calculations with controlling factors.

- l.207-208-209: idem, wind speed, air-sea temperature gradients, tidal amplitude are very briefly presented here in a K600 parametrization paragraph so it is hard to follow. Again, environmental parameters should be presented first before K600. Sub-sections with clear titles in 3.1 section would clearly help as well.

- Table 3: I don't understand well, parametrizations presented in Table 2 were applied to the same datasets (?) over the same area (?) each year between 2015 and 2019 as the present study. It is too bad because, it is not explained by the authors in a clear way in the M&M to help the reader to follow and appreciate measurements and the approach done in the present study. It should be done in the revised version.

If I understand well, the K600 equation obtained from the six days tracer experiment is then applied for each year between 2015 and 2019, am I correct? Which (stations, frequency, sensor, etc…cf. M&M comments above) wind measurements were used for these calculations? How other environmental parameters varied during each year? Variations in K600 values (min-max, …) should be presented and described along other environmental parameters variations.

- l.227: again "in four periods", nothing is explained on this choice by the authors….? Why?

- l.233-237: wind, limited fetch are potential explanations for weak K values indeed, what about other environmental factors such as turbidity, current speeds, depths, rainfall events, heat fluxes…? Authors should discuss this as well as seasonal abiotic and biotic (seagrass growth, phenology, algae, …) effects on gas transfer velocities in Florida Bay since K600 equation presented in the study was obtained during one punctual Spring experiment. Those elements should be discussed in this results and discussion section to go further.

- Fig. 4: see above comments, environmental parameter chronologies should be better described in the MS and linked to K600 analysis after. Here, there are six observation points?

- Fig. 5: it should be modified (graphs a to d separated from graphs e and f) and presented in a clearer and more homogeneous way along the text, there is everything in this figure, similarly to Fig. 4 that should be modified as well (graphs a separated from graphs b, c, d). Graph presentations and associations for each figure should be modified in the revised version.

- Fig. 6: idem and wind speed and K600 values should also be added.

- l.250 (3.2): as written l.255, authors should emphasize this aspect of further tracer or simultaneous air-sea CO2 fluxes and water pCO2 measurements to get more precise k600 parametrizations over seagrasses since (i) dataset here is short (six punctual days) and (ii) relationships between k600 determination with seagrass dynamic (density, phenology, …along with previous works in the area) are not enough shown in the present MS.

- l.256: how many air and water pCO2 values were used or measured? (cf. M&M section, information lacking).

- l.259: what about nighttime period, are there available measurements (pCO2, flux, …) from previous works? It should be discussed. L263-264, assumptions are too speculative and authors should not go too far in their conclusions. Oversaturation periods (respiration, calcification) at night probably exist at their sampling site and additional simultaneous measurements of water/air pCO2 and associated fluxes should be done to draw more precise conclusions (among cited references).

- It could be interesting to better mention in this section other K600 determinations and associated studies among tracer experiments such as floating chamber and particularly atmospheric Eddy Covariance techniques for air-water CO2 flux measurements with simultaneous water pCO2 measurements.

- l.271: cyanobacteria bloom seasonality, what about seagrass as it is the main objective of the paper focusing on seagrass ecosystem?

-l.280: the last sentence is not well formulated and should be modified instead calculating CO2 fluxes from Van Dam et al. 2021 with the 4.5 cm h-1 averaged value authors got in this study and analyzing the difference between both values.

4 Summary:

- l.284-285: again, authors should dampen their conclusion when they compare ("overpredict" word used) obtained K600 values with other from bibliography since (i) they got it over few days in one particular season, (ii) other parametrizations were obtained in very different (and so not comparable) coastal ecosystems (open ocean, rivers, estuaries) and (iii) relationships

with seagrass dynamic and distribution and other environmental parameters are not fully described in the present study.

---

## Author Comment (AC1)

**Revised points for Reviewer 1's comments:**

This is a very nice work and it brings us closer to understanding even more about the air-water CO2 exchange. My only comment/quiestion is, have you consider having longer measurment periods? As I see, there's only a couple of days available. What are the assumptions your are implying in the study if you only calculate transfer velocity for a couple of days? Can it be possible to have a longer period or different sampling periods during the year so you can implement seagrass phenology in the study design?

- Thank you for pointing it out. Even in short periods, we covered wide range of wind speed, and so the derived equation (7) can be used in different seasons if the wind speed is in the range and the seagrass condition is similar. For future study, it is helpful if we measured seagrass density so that we can discuss the relationship between seagrass and $k$. We discussed it as follows.

  "Although the experiment was conducted over a short period of 8 days, our new parameterization, equation (7), fit the observations well; This implies that equation (7) can be applied even in different seasons and years if the wind speed is in the range of 0.12–12 m s$^{-1}$ and seagrass conditions are similar.".

  "Specifically, measuring the seagrass density and conducting dual-tracer experiment simultaneously is needed to relate the $k$ and vegetation distribution"

---

## Author Comment (AC3)

**Revised points for Reviewer 2 (Dr. Pierre Polsenaere)'s comments:**

The submitted manuscript suffers from a lack of information (M&M section with in situ measurements and computations) and organization (especially the result and discussion section) which make difficult to follow and understand the approach and associated results (data set link is not mentioned in the MS sections, only at the end, maybe it can help but the link doesn't work). For instance, seagrass distribution/dynamic and other (than wind, temperature gradient) abiotic environmental parameters should be better described and linked to found gas transfer velocity parametrization. Also, some conclusions should be dampened a bit or reformulated particularly when authors compared their K600 parameterization obtained from a six days dataset in one season, to other different coastal ecosystems cited in the literature. In this way, please see the specific comments below to help in the revision of the different sections and the overall manuscript. I would recommend further revisions to allow the publication of the present paper of Dobashi and T. Ho for the journal EGUsphere.

- We thank the reviewer for the constructive comments. We fixed the link to our data, and it should work now (From the link provided ("https://doi.org/10.5281/zenodo.6730934"), please click "Version Florida 10.5281/zenodo.7087773" in the right column.

Title: It should be specified since very general in the present form.

- OK. we added "— results from a $^{3}$He/SF$_{6}$ tracer release experiment" to the title.

Abstract:

- l.11-12: the general comparison between the present study and other coastal/open oceans is delicate since other environmental factors (than wind speeds) can be involved and controlled k.

- Yes, we agree that the present study's location has different characteristics from other coastal and open oceans. In this sentence, we compare "$k(600)$ at the same speed", which is not "$k(600)$". By comparing "$k(600)$ at the same speed" with other ecosystems, we can say that factors other than wind speeds have effects on $k(600)$. Then, we can start discussing what caused the lower $k(600)$.

- l.13-14 and 16-17: I would dampen this conclusion due to the limited data set and specific studied meadow compared to other seagrass systems worldwide (temperate, subtidal versus intertidal, depth, current, turbidity, rainfall, heat fluxes….).

- The effect of depth and current are weak in this region because the current speed is weak in this region and so the bottom-generated turbulence should be weak. It's not clear why the reviewer thinks turbidity affects gas exchange. There was almost no rain during our survey, and heat flux may be related to the air-sea temperature difference which is discussed in manuscript.

- l.14: other settings? Please specify.

- We replaced "other settings" with "other coastal and offshore regions".

1 Introduction:

- l.21: an important part of seagrass above biomass (refractory matter) is also exported and does not sink to the bottom.

- We discussed it by adding the following phrase.

  "Because the organic carbon produced via photosynthesis easily sinks to the bottom and some of the organic carbon stays in the ocean as a refractory matter, seagrass meadows are expected to be blue carbon sinks that can help mitigate the increase of anthropogenic $CO_2$".

- l.22: Please update the reference adding other more recent works on carbon seagrass storage

- We updated the information by citing Mcleod et al., 2011. The modified sentence is as follows.

  "Seagrasses are estimated to bury 45−190 g C m$^{-2}$ yr$^{-1}$, a significantly higher rate compared to terrestrial forests (0.7−13.1 g C m$^{-2}$ yr$^{-1}$; Mcleod et al., 2011; Duarte et al., 2005)".

- l.25: high methane emissions from seagrass meadow have also been recently shown (see Schorn et al. 2021, PNAS 2022 Vol. 119 No. 9 e2106628119).

- Thank you for the info. We cited the paper you provided. The added sentence is as follows.

  "Schorn et al. (2021) also reported that the seagrasses in the Mediterranean Sea emit 106 μmol m$^{-2}$ d$^{-1}$ methane, mainly from their leaves."

- l.26-27: the link with seagrass here is not clear, please specify.

- They investigated the soil of seagrasses. We added "seagrass meadows in" to the sentence.

- l.33: go further citing other involved parameters (and associated studies) in turbulence control. Also, it lacks in the introduction a review on existing works on gas transfer velocity determination according to (i) coastal ecosystem typology (lagoon, coastal ocean, estuary, seagrass presence or absence, ….) and (ii) methodology, for instance k can also be determined through simultaneous sea-air CO2 fluxes using floating chamber, Eddy Covariance techniques and spatial water pCO2, before focusing on the Florida Bay as done in the next paragraph.

- We added more citations (Ho et al., 1997, 2000) discussing the other parameter of rain. We made a new paragraph to discuss (ii) methodology. For (i), we further introduced the gas transfer experiments in a bay in Hawaii and in an emergent wetland.

New paragraph: "Knowledge of the gas transfer velocity ($k$) is needed to understand the role of seagrass ecosystems in the global carbon cycle, since air-sea $CO_2$ flux is a function of $k$ and the air-sea difference in the partial pressure of $CO_2$ (p$CO_2$). There are several methods to determine $k$ in the field. The $^3$He/SF$_6$ dual tracer technique, which we employed in this study, is a mass balance technique that involves injecting these tracers into the ocean and determining $k$ by measuring the change in the ratio of the two gases with time. The direct flux techniques, such as the eddy covariance method, measure the $CO_2$ flux in the air and $CO_2$ concentration both in the sea and air to derive $k$. $k$ can also be estimated from the transfer velocity of heat by assuming that the gas and heat transfer velocities are related by their diffusivities; however, the estimated gas transfer velocity from heat, $k_H$, have been found to overestimate the actual $k$ (e.g., Atmane et al., 2004)."

Discussing the other parameter of rain: "In the case of rain, rain rate is included in the parameterization because rainfall increases subsurface turbulence and $k$ (Ho et al., 1997a, 2000)."

gas transfer experiments in a bay in Hawaii and emergent wetland : "Ho et al. (2018a) examined $k$ in the Kaneohe Bay in Hawai'i and showed that $k$ can be estimated well by wind speed where the depth is deeper than 10 m."   "Ho et al. (2018b) examined $k$ in emergent wetland where the depth ≤ 1 m, and showed that $k$ can be parameterized from heat flux, rain rate and current velocity there.".

- l.37: please be more specific citing for these studies the used methodology (k, atmospheric CO2 exchanges) among wind speed/gas parametrizations and ecosystem typology as well (presence of seagrasses?).

- We explained how Wanninkhof (1992) determined his parameterization as follows.

    "Zhang and Fischer (2014) determined the air-sea $CO_2$ flux to be 3.93 ± 0.91 mol m$^{-2}$ yr$^{-1}$ in Florida Bay; they used the wind speed/gas exchange parameterization determined from bomb-produced $^{14}$C inventory in the ocean by Wanninkhof (1992)."

2 Material and methods:

l.49: could authors give an idea of seagrass densities in the 2000 km2 and in the specific studied area (Fig. 1). As it stands there is not enough information on spatial seagrass distribution, phenology (carbon stocks and fluxes, …) and concrete relationships done between seagrasses and k parametrization.

- There is a density information 2$^{nd}$ sentence after this sentence. We also added the seagrass density around the study area as follows.

    "Seagrass density varies across the bay, and its standing crop is 0−20 g dry weight m$^{-2}$ in summer around our study area (bottom figure in Fig. 1) (Zieman et al., 1989). The seagrasses in Florida Bay show seasonality, and their standing crop becomes larger in spring and summer and smaller in fall and winter (Zieman et al., 1999).".

- l.60-65: figure caption should be specified and better linked do M&M following sections; it is in between in the submitted MS with cited measured parameters without information on how (sensors, frequencies, …) and when (duration) it was measured. The second zoomed figure should be even more restricted to better see the sampled area. In consequence, we are a bit lost when results and discussion section comes, i.e. parameter origins (sensor used, where, when, frequencies, duration, …).

- We moved Figure 1 to the end of the measurement section. We used a more zoomed figure for the second figure. We added more detailed Material & Method information to the manuscript.

- 2.2.: when and how long were tracers injected? Why?

- We injected 2 days before the first $SF_6$ survey (April 1, 2015) because the generator didn't work the first day when we were out. We added this information:

"We injected $^3$He and $SF_6$ at a ratio of 1:340 into the water at the study location (25.0107°N, 80.692°W; green star in Fig. 1) on 1 April 2015 for 1 minute via a length of diffuser tubing."

2.3.: How many? Where (specify it in Fig. 1)? When?

- We specified it in Fig. 1 and added this information to the manuscript as follows.

"We collected 16 $^3$He samples (~40 mL each) at 26 stations in copper tubes mounted in aluminum channels and sealed at the ends with stainless steel clamps between April 1 and 8 2015 (yellow triangles in Fig. 1)."

"84 discrete $SF_6$ samples were taken at the same stations (yellow triangles in Fig. 1) using 50-mL glass syringes and submerged in water in a cooler until measurement back on shore at the end of each day."

- 2.4., l.89-90: where? When? How? L.92: black square in Fig. 1, how far form the site and why did authors use additional wind data?

- We added this information:

"Hourly tidal amplitude, water surface temperature, and salinity data from the same site (blue dot in Fig. 1) between 2015 and 2019 were obtained from Everglades National Park (https://www.ndbc.noaa.gov/)." and "The tidal amplitude was measured using a digital shaft encoder (WaterLog H331). Water temperature and salinity were measured using multiparameter sondes (Hydrolab Quanta until 5 March 2019; OTT-Hydromet OTT-PLS-C thereafter)".

"Additional wind speeds measured using a sonic anemometer (Vaisala WXT532) at ~3 m above the sea level at 25.07209°N, 80.73511°W (pink square in Fig. 1, 7.4 km away from the blue dot)

between 2015 and 2019 were obtained from Everglades National Park to compare $k$ derived from this study and $k$ estimated from published parameterizations."

L.96-97: a given range for Z0 is given without any explanations on which ecosystem typology in the Florida Bay and how it was measured, please specify. Is the used average Z0 value enough precise for calculations? Maybe I did miss something but why Amorocho and DeVries (1980) equation was not used to compute wind speed data at 10 m?.

- Thank you. It turned out that the (Cornelisen and Thomas, 2009) is not appropriate to our study. We used Amorocho and DeVries (1980) and recalculate. The overall results did not change significantly.

- 2.5., l.100: pCO2 measurements, used frequency? Where? L.106: "specific times", "at regular time intervals" could you specify it? L.107: CO2 standard concentration?

- We added the info as follows

  "We measured the $pCO_2$ along the boat track (red dots in Fig. 1) using an underway system based on the design of Ho et al. (1997b) and incorporating the suggestions from Pierrot et al. (2009)"

  "The interval between measurements was 41 s".

2.7., l.158-159: the last sentence is a little bit awkward, in section 2.7 important testing was done to fit in a precise way coefficient Schmidt numbers especially according to temperature, so why was it not done for salinity in a same way?

- Because the effect of salinity on Sc is not well investigated, as written between the 4[th] and 7[th] sentences in 2.7. We also added the following sentence in Section 2.7.

  "While the effect of temperature on molecular diffusion coefficient is well investigated, the effect of salinity has been the subject of fewer studies."

L150-151: various salinity, you mean 0 and 35, don't you? Various temperatures from 0 to 40°C, could you specify (step, number of values?)? By the way, what are salinity range, mean values in Florida Bay in general and at your sampling area during the study? No result on it in the submitted MS?

- We did not mean 0 and 35 when we used the term "various salinity". The Sc for the salinity of 0 and 35 are just examples. We added the following phrase.

  "The last two columns are the calculated Schmidt number for 20°C, and salinities of 0 and 35 as examples, respectively.".

- We do not need to specify the step and number of values, and our equations can be applied to any temperature between 0 and 40°C.

- Salinity was $40.7 \pm 0.1$ (range=40.4-41.0). We added the temperature and salinity information to section 3.1.

- 3.1, l.183: how many k600 values were taken into account to obtain this average value? That information is lacking in the M&M section and in turn this result value is unclear and one might wonder if this result is comparable to other k600 values found in the bibliography (Fig. 2).

- We have 4 $k(600)$ values since we have 6 $^3$He and $SF_6$ data points and need to calculate the derivative using equation (3). We added the information that we have 6 $^3$He and $SF_6$ data points as follows.

   "We used the mean $^3$He and $SF_6$ concentration for each day to determine $k$, so there are six $^3$He/$SF_6$ data points between April 3 and 8 (Fig. 2f)."

- Fig. 2: five stations were sampled according to M&M l.173. Four observations appear in Fig. 2, could author explain (is it 4 or 5 or anything else)?

- We have 6 $^3$He and $SF_6$ data, 4 $k(600)$ points, as explained in the above comments. M&M173 says 5 stations because we need to neglect the first value to calculate cvRMSE using equation (6).

- We clarified M&M1.173 more as follows.

   "N is the number of stations sampled after the initial sampling (5 for table 2 and 2 for Fig. 5e)"

l.200-201: what did authors mean by "this parametrization" and "K600 between 2015 and 2019"? It is not clear. Could one have further elements or descriptions (sites, measurements, …) at minima instead of only having the two above references?

- "This parameterization" refers to equation 7. We clarified it as follows.

   "$k$ for $CO_2$ at in-situ temperature and salinity between 2015 and 2019 were also calculated using the equation (7) and the previously published parameterizations (Table 3)."

- We were discussing k, not k(600). Thank you for pointing it out. We replaced k(600) with k.

- l.202-203: and using old parametrizations, could we also have those values or at least element of comparisons (%, …)?

- We compared our results and published studies as follows.

   "Annual averaged $k$ ranged between 3.7–4.3 cm h$^{-1}$ in Florida Bay between 2015 and 2019, while published parameterization would yields values of 6.9–11.6 cm h$^{-1}$"

- l.205: I think Results and discussion section should be reorganized, presenting and fully describing first (which has not been done yet) temporal series of measured environmental parameters (grouping Figs. 4 and 6 or at least water pCO2 measurements for instance, Fig. 5) and then K600 descriptions and comparisons with plots (Figs. 2, 3) and tables and relevant controlling environmental factors on k600. As it stands, it is not possible for the reader to see how the ecosystem functioned during these six days experiments before understanding K600 calculations with controlling factors.

- We made new section 3.1 to discuss the environmental settings as follows. As the reviewer suggested above, we discussed the salinity during our experiment here.

"During the experiment, wind direction was predominately from the east, and wind speeds increased towards the latter part of the study period (Fig. 2a). The mean and the standard deviation of the wind speed during the study period was $5.5 \pm 2.0$ m s$^{-1}$ (range=0.12–12 m s$^{-1}$). Mean water temperature showed diurnal pattern with a mean and standard deviation of $26.3 \pm 1.3°C$ (Fig. 2b). The diurnal pattern of the air temperature was weak, as the mean and standard deviation were $25.1 \pm 0.6°C$. The air-sea temperature difference showed diurnal cycles, which was mainly driven by the diurnal cycle of the sea temperature, consistent with observations by Van Dam et al. (2020). Salinity was consistent throughout the study period $(41 \pm 0.1)$ (not shown). The tide consistently showed semidiurnal cycles with an amplitude of $\leq 0.2$ m throughout the study period."

- l.207-208-209: idem, wind speed, air-sea temperature gradients, tidal amplitude are very briefly presented here in a K600 parametrization paragraph so it is hard to follow. Again, environmental parameters should be presented first before K600. Sub-sections with clear titles in 3.1 section would clearly help as well.

- We made new section 3.1 to discuss the environmental settings, as we answered in the above comment.

- Table 3: I don't understand well, parametrizations presented in Table 2 were applied to the same datasets (?) over the same area (?) each year between 2015 and 2019 as the present study. It is too bad because, it is not explained by the authors in a clear way in the M&M to help the reader to follow and appreciate measurements and the approach done in the present study. It should be done in the revised version.

- Table 2 corresponds to section 2.8. Table 3 corresponds to section 2.4. We added an explanation of why we used the wind speed data between 2015 and 2019, which will let readers understand the meaning of table 3. The modified sentence in section 2.4 is as follows.

"Additional wind speeds measured using a sonic anemometer (Vaisala WXT532) at ~3 m above the sea level at 25.07209°N, 80.73511°W (pink square in Fig. 1, 7.4 km away from the blue dot) between 2015 and 2019 were obtained from Everglades National Park to compare $k$ derived from this study and $k$ estimated from published parameterizations".

If I understand well, the K600 equation obtained from the six days tracer experiment is then applied for each year between 2015 and 2019, am I correct? Which (stations, frequency, sensor, etc…cf. M&M comments above) wind measurements were used for these calculations? How other environmental parameters varied during each year? Variations in K600 values (min-max, …) should be presented and described along other environmental parameters variations.

- Yes, that is correct. We wrote the explanation of wind speed measurement in more detail (stations, frequency, sensor, etc. We added annual averages of sea temperature, salinity and tidal amplitude to table 3.

- We added the range of $k$ (min and max) to Table 3.

- l.227: again "in four periods", nothing is explained on this choice by the authors….? Why?

- Because four are the maximum number of periods we can calculate the cvRMSE using the 6 data points. We added the reason why we calculated cvRMSE separately as follows.

  "To investigate the relationship between environmental parameters and the deviation between measured and estimated air-sea gas exchange, we examined the relationship between temperature difference and the deviation between observation and the models by calculating cvRMSE separately in four periods (Fig. 5)."

- l.233-237: wind, limited fetch are potential explanations for weak K values indeed, what about other environmental factors such as turbidity, current speeds, depths, rainfall events, heat fluxes…? Authors should discuss this as well as seasonal abiotic and biotic (seagrass growth, phenology, algae, …) effects on gas transfer velocities in Florida Bay since K600 equation presented in the study was obtained during one punctual Spring experiment. Those elements should be discussed in this results and discussion section to go further.

- We do not understand why turbidity affect gas exchange.

- The impact of current speed and depth is weak since the current speed is weak and thus the bottom-generated turbulence is weak. We observed almost no rain during the experiment (rain amount was 0 during our observation period except there was one hour with 0.5 inch per hour, which is small). We discussed the seasonality of seagrass since the wave attenuation by the seagrass depends on the seagrass density as follows.

  "The seagrasses in Florida Bay show seasonality, and their standing crop becomes larger in spring and summer and smaller in fall and winter (Zieman et al., 1999)."

  "Although the experiment was conducted over a short period of 8 days, our new parameterization, equation (7), fit the observations well; This implies that equation (7) can be applied even in different

- The relationship between heat flux and gas transfer velocity was investigated by Van dam et al., 2020. We are discussing their study and our study in the previous paragraph.

- Fig. 4: see above comments, environmental parameter chronologies should be better described in the MS and linked to K600 analysis after. Here, there are six observation points?

- Yes, we have 6 observational plots since we have 6 pairs of $^3$He and SF$_6$ data. We described more about the material and method as suggested in the other comments.

Fig. 5: it should be modified (graphs a to d separated from graphs e and f) and presented in a clearer and more homogeneous way along the text, there is everything in this figure, similarly to Fig. 4 that should be modified as well (graphs a separated from graphs b, c, d). Graph presentations and associations for each figure should be modified in the revised version.

- We added the notation of "Period 1"~"Period 4" to the Figures 5(a)~(d) so that readers can understand the relationship between (a)-(d) and (e)-(f). We did not separate figures, but now readers can see when is period 1 from figures (a)-(d) easily.

- Fig. 6: idem and wind speed and K600 values should also be added.

- We added the time series of $k$ to Fig. 2e. wind is already shown in Fig. 2a.

- l.250 (3.2): as written l.255, authors should emphasize this aspect of further tracer or simultaneous air-sea CO2 fluxes and water pCO2 measurements to get more precise k600 parametrizations over seagrasses since (i) dataset here is short (six punctual days) and (ii) relationships between k600 determination with seagrass dynamic (density, phenology, …along with previous works in the area) are not enough shown in the present MS.

- OK. We discussed the future study as follows.

  "Specifically, measuring the seagrass density and conducting dual-tracer experiment simultaneously is needed to relate the $k$ and vegetation distribution."

- We also discussed the implication of the short experiment as follows.

  "Although the experiment was conducted over a short period of 8 days, our new parameterization, equation (7), fit the observations well; This implies that equation (7) can be applied even in different seasons and years if the wind speed is in the range of 0.12–12 m s$^{-1}$ and seagrass conditions are similar.".

- l.256: how many air and water pCO2 values were used or measured? (cf. M&M section, information

lacking).

- We added the information in method section as follows.

    "In total, 1,261 and 13 $xCO_2$ data were taken from the water and air, respectively."

- l.259: what about nighttime period, are there available measurements (pCO2, flux, …) from previous works? It should be discussed. L263-264, assumptions are too speculative and authors should not go too far in their conclusions. Oversaturation periods (respiration, calcification) at night probably exist at their sampling site and additional simultaneous measurements of water/air pCO2 and associated fluxes should be done to draw more precise conclusions (among cited references).

- To investigate $pCO_2$ during the nighttime, we referred to the $pCO_2$ data from NOAA near the observational station (15 km away from our station). At NOAA's station, the $CO_2$ flux was always negative and amplitude of diurnal $fCO_{2water}$ was 25–53 μatm between April 3 and 8, 2015. We modified the sentence as follows.

    "Although we did not conduct $pCO_2$ measurement during the night and so the calculated value is biased toward daytime, the daily averaged $pCO_{2water}$ and $CO_2$ flux during the whole observation period would still be lower than $pCO_{2air}$ and negative, respectively, considering that the observed $pCO_2$ was as low as 228 μatm and the $CO_2$ flux at the NOAA station (aqua diamond in Fig. 1) was always negative with diurnal $fCO_2$ amplitude of 25–53 μatm between April 3 and 8, 2015."

- It could be interesting to better mention in this section other K600 determinations and associated studies among tracer experiments such as floating chamber and particularly atmospheric Eddy Covariance techniques for air-water CO2 flux measurements with simultaneous water pCO2 measurements.

- We discussed the paper from Van Dam et al. 2020 in the previous section, which determined $k$ from heat flux. As far as we know, there are no air-sea gas exchange measurements using the floating chamber or eddy covariance technique in Florida Bay.

- l.271: cyanobacteria bloom seasonality, what about seagrass as it is the main objective of the paper focusing on seagrass ecosystem?

- That is a good point. Thank you. We discussed it as follows.

    "The seasonality of seagrasses may also contribute to the seasonality of $pCO_2$ and $CO_2$ flux, as its productivity also shows seasonality (higher in spring and summer and lower in fall and winter) (Zieman et al., 1999).".

-l.280: the last sentence is not well formulated and should be modified instead calculating CO2 fluxes from Van Dam et al. 2021 with the 4.5 cm h-1 averaged value authors got in this study and analyzing the difference

between both values.

- Ok. We re-calculated the excess $CO_2$ using our equation (7). The modified sentence is as follows.

  "Van Dam et al. (2021) also calculated the excess $CO_2$, which is the $CO_2$ concentration difference between water and air to achieve the annual $CO_2$ flux of 6.1–7.0 mol m$^{-2}$ year$^{-1}$, in Florida Bay to be between 5.2 and 6.0–7.0 μmol kg$^{-1}$, using a mean $k$ of 11.7 cm h$^{-1}$; we recalculated the excess $CO_2$ to be between 14 and 16 μmol kg$^{-1}$ using the $k$ of 4.3 cm h$^{-1}$, which is parameterized from this study (Table 3). The recalculated excess $CO_2$ almost double their calculation of 5.2–6.0 μmol kg$^{-1}$ and hence require more $CO_2$ input".

4 Summary:

- l.284-285: again, authors should dampen their conclusion when they compare ("overpredict" word used) obtained K600 values with other from bibliography since (i) they got it over few days in one particular season, (ii) other parametrizations were obtained in very different (and so not comparable) coastal ecosystems (open ocean, rivers, estuaries) and (iii) relationships with seagrass dynamic and distribution and other environmental parameters are not fully described in the present study.

- It is true that the parameterization from published studies overpredict $k$ in this area, and it is one of our main results. Even though it was a short experiment, we covered a wide range of wind speeds. We added the discussion about the effect of rain and bottom-generated turbulence as you commented above.

---

## Referee Report (RR1)

**Review on the egusphere-2022-525-manuscript-version2 (revised MS version) from Ryo Dobashi and David T. Ho**

I thank the authors for their responses to my previous comments and the associated revised version of their MS. Most of my comments have been addressed and the revised version has been specified and improved. However, before publication in EGUsphere, significant improvement still need to be done in my opinion to improve the MS with regards to 1) the English language through an official English editing service or other options and 2) its scientific organization both for the Methods and the Results and Discussion sections. In the Methods section, there are 8 sub-sections that could easily be grouped in tracer measurements, environmental measurements (environmental variables, pCO2, etc.), tracers, Sc number, k calculations/modeling. The same effort has to be done for the Results and Discussion section with homogeneous sections and associated paragraphs, explicit titles, etc. Here are as well below, specific comments that need to be addressed to help authors improve the MS.

**Introduction**

l.18: Why has parameterization changed between the submitted and the revised MS?

l.28-32: Reformulate the whole paragraph saying first seagrasses can also emit GES (CO2, CH4) and then giving the two examples for CO2 emissions from CaCO3 production and CH4 emissions as well. In the Howard et al. (2017) study, it is not clear as it is written in the revised MS the link between the fact there is more IC than OC and the systems are CO2 sources?

l.40: For k estimations from simultaneous EC and $pCO_2$ measurements, you can cite this work (though no obligation at all) to support your idea: Polsenaere P., Deborde J., Detandt G., Vidal L.O., Pérez M.A.P., Marieu M., and Abril G. (2013) Thermal enhancement of gas transfer velocity of $CO_2$ in an Amazon floodplain lake revealed by Eddy Covariance measurements. *Geophysical Research Letters,* 40, 1-7, doi:10.1002/grl.50291. Idem in l.50 for heat flux control on K for floodplain in Amazonia.

l.56: was instead of is

**Methods**

Pink squares are illegible, please change colour

Table 1 in Supplementary Material?

**Results and Discussion**

l.314 See Abril et al. (2009) ECSS 83, 342-348 to understand how and why turbidity can affect gas exchange (authors response to previous comment)

l.316 "seagrass conditions are similar", please specify it in the revised MS. In consequence, I still (last previous comments) think conclusions on K relationships with seagrass dynamic and distribution and extension to other seagrass systems can't completely done here.

---

## Author Response (AR2)

MS No.: egusphere-2022-525

Air-sea gas exchange in a seagrass ecosystem— results from a $^3$He/SF$_6$ tracer release experiment

Ryo Dobashi and David T. Ho

Extended revision due date: 27 Jan 2023

**Revised points for Dr. Pierre Polsenaere's comments:**

I thank the authors for their responses to my previous comments and the associated revised version of their MS. Most of my comments have been addressed and the revised version has been specified and improved. However, before publication in EGUsphere, significant improvement still need to be done in my opinion to improve the MS with regards to 1) the English language through an official English editing service or other options and 2) its scientific organization both for the Methods and the Results and Discussion sections. In the Methods section, there are 8 sub-sections that could easily be grouped in tracer measurements, environmental measurements (environmental variables, pCO2, etc.), tracers, Sc number, k calculations/modeling. The same effort has to be done for the Results and Discussion section with homogeneous sections and associated paragraphs, explicit titles, etc. Here are as well below, specific comments that need to be addressed to help authors improve the MS.

- We appreciate your constructive comments again.

- First, we modified figures so that readers can see them easily. For Fig. 1, we deleted the small map at upper left corner showing the location of Florida Bay. We made the bottom map bigger. We changed the shapes of Figs 3 and 4 to squares.

  For Fig. 5, we made (a)-(d) bigger. For Fig. 6, we only plotted the period when we were at measurement site so that the detailed $CO_2$ flux variability can be seen. We also calculated daily $CO_2$ flux in addition to daytime $CO_2$ flux by assuming diurnal amplitude in $CO_2$ difference is small. We discussed the variability of $CO_2$ flux as follows (section 3.3 2$^{nd}$ paragraph).

  "The calculated daytime $CO_2$ flux using the measured $p$CO$_2$ difference and modeled $k$ in this study (Black solid line in Fig. 2e) was $–5.3 \pm 3.0$ mmol m$^{-2}$ day$^{-1}$ (negative value denotes CO$_2$ flux from the air to the water) (Fig. 6b). The $CO_2$ flux varied both within a day and between days mainly due to the variability in $k$ (Note that $k$(600) in Fig. 2e is filtered with 25 minutes running average). Diurnal fCO$_{2water}$ amplitude at the NOAA station (cyan diamond in Fig. 1) between 3 and 8 April 2015 was as small as 25–53 µatm, and so we calculated daily $CO_2$ flux by assuming $CO_2$ difference between air and water during the night is the mean daytime $CO_2$ difference. The calculated daily $CO_2$ flux was $–7.0 \pm 3.5$ mmol m$^{-2}$ day$^{-1}$, which was higher than daytime $CO_2$ flux because wind speed was higher during the night."

  Since summary was too short and not so clear, we modified several sentences as follows.

"Air-sea gas exchange was investigated in a seagrass ecosystem in South Florida, USA, using the $^3$He and $SF_6$ dual tracer technique. The gas transfer velocity was lower than that in other coastal areas and open oceans, and commonly-used wind speed/gas exchange parameterizations overpredict the gas transfer velocities, especially when wind speeds were relatively high (> 7 m s$^{-1}$). A new wind speed/gas exchange parameterization was proposed ($k(600) = 0.143u_{10}^2$), which was able to predict the observed gas transfer velocities significantly better than existing parameterizations. This result suggests that wind is the dominant factor controlling gas exchange in the studied seagrass ecosystem, but the lower gas transfer velocity at a given wind speed was due to limited wind fetch in the study area and wave attenuation by seagrass. To assess the wider applicability of the proposed wind speed/gas exchange parameterization, more tracer release experiments are needed at similar inland ecosystems"

We reply to your specific comments below.

- We reorganized the title and paragraph as you pointed out. The reorganized titles are listed below.

  1 Introduction

  2 Methods

      2.1 Study site

      2.2 Tracer injection and measurement

      2.3 Measurements of environmental variables

      2.4. Gas transfer velocity calculation and $^3$He/$SF_6$ ratio modeling

      2.5 Calculation of $Sc$ number

   3. Results and discussion

      3.1 Environmental parameters

      3.2 Gas transfer velocity in Florida Bay and assessment of published parameterization

      3.3 Implications for biogeochemistry

   4. Summary

Introduction:

l.18: Why has parameterization changed between the submitted and the revised MS?.

- It is because the equation to derive wind speed at 10 m height ($u_{10}$) was changed. In the previous manuscript, we calculated $u_{10}$ by using roughness length from Cornelisen and Thomas (2009), but it turned out that the roughness length from the paper was not appropriate to our study. We are

now using the equation from Amorocho and DeVries (1980) as you suggested in the present review.

l.28-32: Reformulate the whole paragraph saying first seagrasses can also emit GES (CO2, CH4) and then giving the two examples for CO2 emissions from CaCO3 production and CH4 emissions as well. In the Howard et al. (2017) study, it is not clear as it is written in the revised MS the link between the fact there is more IC than OC and the systems are CO2 sources?

- We changed the sentences as you suggested. The modified sentence is as follows(1st paragraph in section 1).

  "However, recently, the role of seagrasses in the global carbon cycle has been revisited, as carbon emissions from seagrasses were found to be large (Howard et al., 2017; Van dam et al., 2021; Schorn et al., 2021). Howard et al. (2017) examined the stock of organic and inorganic carbon in the soil of seagrass meadows in Florida Bay and southeastern Brazil, and found that the soils in both regions have more inorganic than organic carbon. They suggested that both regions are sources of $CO_2$ to the atmosphere by assuming 0.6 mol of $CO_2$ is produced when 1 mol of $CaCO_3$ is produced. Schorn et al. (2021) reported that the seagrasses in the Mediterranean Sea emit 106 $\mu mol\ m^{-2}\ d^{-1}$ methane, mainly from their leaves.".

l.40: For k estimations from simultaneous EC and pCO2 measurements, you can cite this work (though no obligation at all) to support your idea: Polsenaere P., Deborde J., Detandt G., Vidal L.O., Pérez M.A.P., Marieu M., and Abril G. (2013) Thermal enhancement of gas transfer velocity of CO2 in an Amazon floodplain lake revealed by Eddy Covariance measurements. Geophysical Research Letters, 40, 1-7, doi:10.1002/grl.50291. Idem in l.50 for heat flux control on K for floodplain in Amazonia.

- Thank you for the information. We cited a paper introducing direct flux measurements alternatively (McGillis et al. 2001) when we introduce various $k$ estimation methods (section 1, 2nd paragraph). The added sentence is as follows.

  "The direct flux techniques, such as the eddy covariance method, measure the $CO_2$ flux in the air and $CO_2$ concentration both in the sea and air to derive $k$ (McGillis et al. 2001)."

l.56: was instead of is

- We changed from "is" to "was".

Methods:

Pink squares are illegible, please change colour.

- We did not change the color, but we made the plot bigger. We also surrounded the pink square by a black line.

Table 1 in Supplementary Material?

- We think Table 1 provides valuable information to some readers, so we will not move this table to the supplementary material.

Results and discussions:

l.314 See Abril et al. (2009) ECSS 83, 342-348 to understand how and why turbidity can affect gas exchange (authors response to previous comment)

- Thank you for letting us know the paper. This paper found that air-sea gas exchange is suppressed when turbidity has high concentration. However, the water was clear when we conducted our measurements and so we think the effect is minor.

l.316 "seagrass conditions are similar", please specify it in the revised MS. In consequence, I still (last previous comments) think conclusions on K relationships with seagrass dynamic and distribution and extension to other seagrass systems can't completely done here.

- Relating the vegetation and gas transfer velocity will be future study, but we consider that the proposed equation can be used on regions where seagrass density is similar to our study region. We specified "seagrass conditions are similar" by adding the information of seagrass density as follows.

  "Although the experiment was conducted over a short period of 8 days, our new parameterization, equation (7), fit the observations well; This implies that equation (7) can be applied even in different seasons and years if the wind speed is in the range of 0.12–12 m s$^{-1}$ and seagrass conditions are similar (dominant seagrass of _Thalassia testudinum_ has 63.6 (range=0–215) g dry weight m$^{-2}$ standing crop in Florida Bay (Zieman et al., 1989)).".

---

## Author Response (AR3)

MS No.: egusphere-2022-525

Air-sea gas exchange in a seagrass ecosystem— results from a $^{3}$He/SF$_6$ tracer release experiment

Ryo Dobashi and David T. Ho

Extended revision due date: 08 Feb 2023

**Revised points for Dr. Peter Landschützer's comments:**

many thanks for revising the manuscript. Based on my reading, I am now happy to accept your manuscript for publication in BG. I have however indicated technical corrections in my decision report. What I would like you to consider - upon uploading the final files - is to improve readability of the figures for visually impaired readers. This concerns figures 3, 5 and 2e, where color lines are close to each other. If possible, adding symbols or dashes would help the reader next to the difference in color.

- We appreciate your constructive comment. We changed some lines from solid to dash, dot, or dash-dotted. For Fig. 2e, 25 min running average was changed to 65 min running average so that the lines can be more easily distinguished from each other. We changed the explanation of 2$^{nd}$ paragraph in Section 3.3 as follows, but there is no effect on the overall story.

  "(Note that $k(600)$ in Fig. 2e is filtered with 65 minutes running average)"

  We also removed the placeholders "Disclaimer" and "Review statement" since we do not have anything to fill in there.